

**Ozone pollution around a coastal region of South China Sea: Interaction between marine and**
**continental air**
Hao Wang[1#], Xiaopu Lyu[1#], Hai Guo[*, 1], Yu Wang[1], Shichun Zou[2], Zhenhao Ling[3], Xinming
Wang[4], Fei Jiang[**, 5], Yangzong Zeren[1], Wenzhuo Pan[1], Xiaobo Huang[6], Jin Shen[7]
[1] Air Quality Studies, Department of Civil and Environmental Engineering, Hong Kong
Polytechnic University, Hong Kong
[2] School of Marine Sciences, Sun Yat-sen University, China
[3] School of Atmospheric Sciences, Sun Yat-sen University, China
[4] Guangzhou Institute of Geochemistry, Chinese Academy of Sciences, Guangzhou, China
[5] International Institute for Earth System Science, Nanjing University, China
[6] Shenzhen Academy of Environmental Sciences, Shenzhen, China
[7] State Key Laboratory of Regional Air Quality Monitoring, Guangdong Key Laboratory of
Secondary Air Pollution Research, Guangdong Environmental Monitoring Center,
Guangzhou, China
*Corresponding to:* * H. Guo (ceguohai@polyu.edu.hk) and ** F. Jiang (jiangf@nju.edu.cn)
[#] These authors contributed equally to this work.



**Abstract**
Marine atmosphere is usually considered to be a clean environment, while this study indicates
that the near-coast waters of South China Sea (SCS) suffered from even worse air quality
than coastal cities. The analyses were based on concurrent field measurements of target air
pollutants and meteorological parameters conducted at a suburban site (TC) and a nearby
marine site (WS) from August to November 2013. The observations showed that the levels of
primary air pollutants were significantly lower at WS than those at TC, while ozone ($O_3$)
value was greater at WS. Higher $O_3$ levels at WS were attributed to the weaker NO titration
and higher $O_3$ production rate because of stronger oxidative capacity of the atmosphere.
However, $O_3$ episodes were concurrently observed at both sites under certain meteorological
conditions, such as tropical cyclones, continental anticyclones and sea-land breezes (SLBs).
Driven by these synoptic systems and mesoscale recirculations, the interaction between
continental and marine air masses had profoundly changed the atmospheric composition and
subsequently influenced the formation and redistribution of $O_3$ in the coastal areas. When
continental air intruded into marine atmosphere, the $O_3$ pollution was magnified over SCS,
and the elevated $O_3$ (>100 ppbv) could overspread the sea boundary layer ~8 times the area of
Hong Kong. In some cases, the exaggerated $O_3$ pollution over the SCS was re-circulated to
the coastal inshore by sea breeze, leading to even aggravating $O_3$ pollution in coastal cities.
The findings are applicable to similar mesoscale environments around the world where the
maritime atmosphere is potentially influenced by severe continental air pollution.
**Key words**: Continental air pollution; Maritime atmosphere; Mesoscale recirculation; Ozone
photochemistry





## 1 Introduction

Ozone ($O_3$) plays a central role in photochemical oxidation processes in the troposphere via direct reaction, photolysis and the subsequent reactions to produce the hydroxyl radical (Monks et al., 2015; Seinfeld and Pandis, 2016). As a strong oxidant, $O_3$ at surface level is recognized to be a threat to human health (WHO, 2003; Bell et al., 2007) and has a detrimental impact on vegetation (Fowler et al., 2009) and built infrastructure (Kumar and Imam, 2013). Tropospheric $O_3$ is also the third most important greenhouse gas (IPCC, 2014) and is referred to a short-lived climate pollutant (Shindell et al., 2012).

To mitigate $O_3$ pollution in the troposphere, tremendous efforts from both scientific and regulatory communities have been made since three decades ago (NRC, 1991; NARSTO, 2000; Monks et al., 2015). The $O_3$ levels started to decrease at many locations, such as Jungfraujoch in Switzerland, Zugspitze in Germany, Mace Head in Ireland, as well as parts of California and eastern US (Lefohn et al., 2010; Cui et al., 2011; Parrish et al., 2012; Lin et al., 2017). However, increasing studies showed that surface $O_3$ was elevated rapidly in East Asia in the last decade (Ding et al., 2008; Xu et al., 2008; Parrish et al., 2012; Zhang et al., 2014; Lin et al., 2017; Wang et al., 2017a). For example, the observational data revealed that regional $O_3$ concentrations increased at a rate of 0.86 ppbv $yr^{-1}$ in Pearl River Delta (PRD) from 2006 to 2011 (Li et al., 2014), and at a rate of 0.56 ppbv $yr^{-1}$ in Hong Kong from 2005 to 2014 (Wang et al., 2017a).

Hong Kong and the adjacent PRD is the most industrialized region along the coast of South China Sea (SCS), and is suffering from serious $O_3$ pollution (Zheng et al., 2010; Derwent et al., 2013; Ling et al., 2013). Numerous studies demonstrated that in addition to long-range transport (Chan, 2002; Guo et al., 2009; Wang et al., 2009) and local photochemical production (Ding et al., 2013a), tropical cyclones and mesoscale circulations are conducive to the occurrence of high $O_3$ events (Yin, 2004; Huang et al., 2005; Yang et al., 2012; Jiang et al., 2015; Wei et al., 2016). In a number of studies, tropical cyclone has been considered as the most conducive driver to the occurrence of $O_3$ episodes in Hong Kong (Yin, 2004; Ling et al., 2013) for it generally causes peripheral subsidence, stagnation air and inversion layer, which favor the production and accumulation of $O_3$.

Mesoscale circulations (*e.g.,* sea-land breezes (SLB) and mountain–valley breezes) also play important roles in $O_3$ distribution and transport in the coastal cities like Hong Kong with



complex topography and land cover ( Liu and Chan, 2002; Ding et al., 2004; Lu et al., 2009a;
Guo et al., 2013). For instance, Guo et al. (2013) demonstrated that upslope winds brought
pollutants including $O_3$ from low-lying areas to the peak of Mt Tai Mo Shan (957 m a.s.l.) in
Hong Kong. Ding et al. (2004) simulated a multi-day SLB related $O_3$ episode and discussed
the influence of SLB circulation on the transport of oxidant precursors, the residence time
and re-entry of photochemical compounds. Lu et al. (2010) simulated the SLB in the
2003/2004 winter and revealed that the urbanization of Shenzhen might significantly enhance
the sea breeze to the west of Hong Kong in the early afternoon, which worsened the local air
pollution.
Both coastal human activities and marine atmospheric cyclic behavior can significantly affect
the air pollution level in coastal urban environments (Adame et al., 2010; Velchev et al.,
2011). Exploring SLBs provides an important way to understand the interaction between
continental air and marine atmosphere which has long been a focus of coastal air quality,
global tropospheric chemistry and climate change research. Surprisingly, few studies
investigated SLBs in Hong Kong though about 70 SLB days per year on average were
observed in Hong Kong and the PRD region (Zhang and Zhang, 1997). Therefore, the
association between mesoscale recirculation and air pollutants over the SCS and subtropical
continental region is still not well established, which seriously limits our understanding on
the interplay of continental and marine air masses in this region. Furthermore, previous $O_3$
studies carried out in this region neither paid enough attention to the variations of volatile
organic compounds (VOCs, one important group of $O_3$ precursors) nor established any field
measurements on an island, an ideal site for observation of marine air mass with less
interference from local emissions, for understanding the $O_3$ pollution around the coastal
region of the SCS (Parrish et al., 1998). So far, only a handful of studies deeply evaluated the
chemical characteristics of air masses under various synoptic systems (Wang et al., 2005;
Guo et al., 2009, 2013).
This study aimed to comprehensively characterize interaction between continental
anthropogenic emissions and marine atmosphere over a coastal region of the SCS by
concurrent measurements and in-depth analysis of air pollutants at a marine site over SCS
and a suburban site in Hong Kong. Firstly, the spatial and temporal variations of
measurements were described to give an overall picture of the campaign, as well as to
directly evaluate how continental outflows polluted the marine atmosphere over the SCS.





After that, the chemical and meteorological characteristics of air masses associated with high
$O_3$ concentrations were explored. Finally, the interplay between the maritime and continental
air masses and its influence on regional air quality were discussed.
**2 Methodology**
**2.1 Sampling sites**
Field measurements were carried out concurrently at a suburban site and a marine site over
SCS (Figure 1). The suburban Tung Chung (TC, 22.29 °N, 113.94 °E) site, part of the Hong
Kong Environmental Protection Department (HKEPD) air quality monitoring network, is
located in southwestern Hong Kong, about 3 km south of the Hong Kong International
Airport at Chek Lap Kok with Hong Kong urban center about 20 km to the southwest and
Macau 38 km to the northeast. It is a newly-developed residential town adjacent to the busy
highway and railway lines. The sampling instruments were installed on the rooftop of a
building with a height of 27.5 m a.s.l. More detailed description of the TC site can be found
in our previous publications (Cheng et al., 2010a; Jiang et al., 2010).
The marine site, Wan Shan island (WS, 21.93 °N, 113.73 °E), is located 40 km southeast of
Zhuhai, and is bounded to the north by the Pearl River Estuary, with a straight distance of
about 44 km to TC. WS has an area of 8.1 $km^2$ and a population of about 3,000 with sparse
anthropogenic emissions at the island. The isolated island features a sub-tropical maritime
climate. The measurement site was set up on the rooftop of the National Marine
Environmental Monitoring Station with a height of about 65 m a.s.l.
High $O_3$ mixing ratios are frequently observed in Hong Kong in late summer and autumn
(Ling et al., 2013) when the northeast monsoon prevails. During this period, WS is right in
the downwind direction of TC, which facilitates the study of the interaction between the
inland pollution and the marine environment.





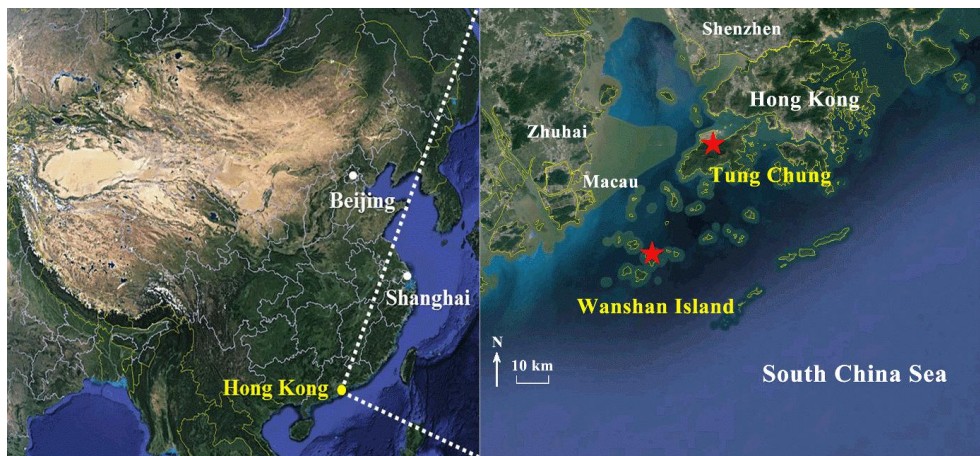

**Figure 1.** Locations of the sampling sites  (red stars) and the surrounding environment.
**2.2 Measurement techniques**
*2.2.1 Measurements of trace gases and meteorological parameters*
The sampling campaign was conducted from 10 Aug. to 21 Nov. across late summer and
autumn in 2013. At WS, trace gases (*i.e.*, $NO_x$, $O_3$, $SO_2$ and CO) were continuously monitored
with a time resolution of 1 minute. $NO$-$NO_2$-$NO_x$ was measured using a
chemiluminescence analyzer (*Thermo Environmental Instruments (TEI), Model 42i*) with a
range of 0-200 ppbv and a lower detection limit of 0.40 ppbv. $O_3$ was monitored with a
commercial UV photometric analyzer (*TEI, Model 49i*) with a range of 0-0.050 to 200 ppm
and a lower detection limit of 1.0 ppbv. $SO_2$ was measured using a pulsed UV fluorescence
approach (*TEI, Model 43S*). CO was measured by a gas filter correlation, non-dispersive
infrared analyzer (*API, Model 300*) with a heated catalytic scrubber to convert CO to $CO_2$
for baseline determination. Quality assurance and control procedures (*e.g.*, instrumental
maintenance and calibration) for these devices have been described elsewhere (Guo et al.,
2009, 2013). Meteorological parameters, including temperature, relative humidity, solar
radiation, wind speed and wind direction, were routinely monitored by a weather station
(Vantage Pro 2 plus, Davis Instruments) with a time resolution of 5 minutes. At TC, hourly
data of the aforementioned trace gases and meteorological parameters were obtained from the



HKEPD (http://epic.epd.gov.hk/ca/uid/airdata). Detailed information of the quality assurance
and control protocols is available in the HKEPD report (HKEPD, 2015).
*2.2.2 Sampling and analysis of VOCs*
Concurrent VOC samples (*i.e.,* non-methane hydrocarbons (NMHCs) and carbonyls) were
collected on 21 selected days (including both non-$O_3$ episodes and $O_3$ episodes) at both
sites. These days were selected on the basis of weather prediction and meteorological data
analysis for potentially high and low $O_3$ days. An $O_3$ episode day is the day when the peak
one-hour averaged $O_3$ mixing ratio exceeds 100 ppbv (Level II of China National Ambient
Air Quality Standard). Please refer to our previous publication for details of this method (Guo
et al., 2009).
The whole-air samples of NMHCs were collected using 2-L electro-polished stainless steel
canisters. The canisters were cleaned, conditioned and evacuated before being used for
sampling. A metal bellows pump was used to fill up the canisters with sample air over one-
hour integration (with a flow restrictor) to a pressure of 40 psi. Seven one-hour VOC
samples (every two hours during 7:00 – 19:00 inclusive) were collected simultaneously at
each site. Intensive VOC sampling was also carried out at WS in selected seven days (*i.e.*, 3, 4, 9
and 22-25 October) with eleven one-hour samples (every two hours during 1:00 – 22:00
inclusive). Totally, 311 valid VOC samples (144 at TC and 167 at WS) were collected in
addition to about 5% field blanks and 5% parallel samples for quality assurance purpose. The
speciation and abundance of 59 $C_2$-$C_{11}$ NMHCs in the canisters were determined by a Model
7100 preconcentrator (Entech Instruments Inc., California, USA) coupled with an Agilent
5973N gas chromatography-mass selective detector/flame ionization detector (GC-MSD/FID,
Agilent Technologies, USA). The detection limit of NMHCs was 3 pptv with a measurement
precision of 2-5%, and a measurement accuracy of 5%. Detailed information of the analysis
system and quality control and quality assurance for VOC samples can be found elsewhere
(Simpson et al., 2010).
Carbonyl samples were collected using silica gel filled cartridges impregnated with acidified
2,4-dinitrophenylhydrazine (DNPH). Air samples were drawn through the cartridge at a flow
rate of 0.8–0.9 L min$^{-1}$ for 2 hours; the flow rate through the cartridges was monitored with a
rotameter which was calibrated before and after each sampling. An $O_3$ scrubber was
connected to the inlet of the DNPH–silica gel cartridge to prevent interference from $O_3$. In



total, 227 carbonyl samples (124 at TC and 103 at WS) were collected with 5 and 6 samples
per non-$O_3$ and $O_3$ episode day (every two hours during 7:00 - 18:00 inclusive), respectively.
All cartridges were stored in a refrigerator at 4 ℃ after sampling. The sampled carbonyl
cartridges were eluted slowly with <5 ml of acetonitrile in the direction opposite to sampling
flow into a 5-ml brown volumetric flask, followed by adding acetonitrile to a constant
volume of 5 ml. A 20-μl aliquot was injected into the high performance liquid
chromatography (HPLC) system through an auto-sampler. The operating conditions of the
HPLC are shown in Table S1. Typically, $C_1$–$C_9$ carbonyl compounds were measured
efficiently with a detection limit of ~0.2 ppbv.
**2.3 Observation-based model (OBM)**
A photochemical box model coupled with the Master Chemical Mechanism v3.2 (PBM-
MCM) was applied to simulate the $O_3$ production at WS and TC for the VOC sampling days.
The PBM-MCM model is a zero-dimension photochemical box model combined with a near
explicit chemical mechanism consisting of 5,900 species and 16,500 reactions, which fully
describes the mechanisms of homogeneous reactions in the atmosphere (Jenkin et al., 1997;
Jenkin et al., 2003; Saunders et al., 2003). The simulation was constrained by hourly data of
meteorological parameters (*i.e.,* temperature and relative humidity) and air pollutants (NO,
$NO_2$, CO, $SO_2$ and 51 measured VOCs). Since the sampling interval was two hours for each
sample, cubic spline interpolation was used to derive VOC concentrations at each hour for
modeling purpose. Please see our previous publication for details (Wang et al., 2017a). It is
noteworthy that the atmospheric physical processes (*i.e.*, vertical and horizontal transport)
were not considered in this model. The PBM-MCM model has been successfully applied in
previous studies (Cheng et al., 2010b; Lam et al., 2013; Ling et al., 2014). Details of the
model construction can be found in Saunders et al. (2003) and Lam et al. (2013).
**2.4 WRF-CMAQ simulation and backward particle release model**
In this study, the Weather Research and Forecasting (WRF v3.7.1) model (Skamarock et al.,
2008) was used to simulate vertical and horizontal wind fields for various weather systems
observed in this campaign, and then provided meteorological parameters required by U.S.
EPA Community Multiscale Air Quality (CMAQ v4.7.1) model (www.epa.gov/cmaq).
CMAQ is a three-dimensional Eulerian atmospheric chemistry and transport modeling system,
which includes complex physical and chemical processes, such as physical transport and
diffuse, gas and aqueous chemical transformation, and so on; and it can treat multiple





pollutants simultaneously from local to continental scales. A domain system composed of
four nested grids (81, 27, 9, 3 km) was adopted to better suit the simulation of mesoscale
weather systems, as shown in Figure S1. The domain with finest resolution (3 km) covers the
Pearl River Estuary region. Vertically, there were 31 sigma levels for all domains, with the
model top fixed at 50 hPa. The major selected physical schemes invoked in WRF and
chemical mechanisms used in CMAQ are shown in Table S2. The input meteorological data
was made using NCEP FNL (final) data with a horizontal resolution of $1^{o} \times 1^{o}$
(https://rda.ucar.edu/). In addition, the geographical data were obtained from the Research
Data Archive of National Center for Atmospheric Research (NCAR)
(http://www2.mmm.ucar.edu/wrf/users /downloads.html). The emission inventories (EI) used
in this study included the 2000-based Regional Emission Inventory in ASia (REAS)
(Kurokawa et al., 2013) and the 2010-based Multi-resolution Emission Inventory for China
(MEIC) (He, 2012), both of which were processed by the Sparse Matrix Operating Kernel
Emission (SMOKE) model. The biogenic emissions were calculated by the Model of
Emissions of Gases and Aerosols from Nature (MEGAN) (Guenther, 2006). The WRF
modelling mainly focused on $O_3$ episodes with an additional 24hrs' preceding run as spin-up
for each episode, and the integration was conducted separately. In addition, the
spatiotemporal patterns of CO and $O_3$ were simulated by WRF-CMAQ during two $O_3$
episodes (see section 3.4). Table S3 gives the index of agreements (IOAs) between the
simulated and observed meteorological parameters and air pollutants. Within the range of 0 –
1, higher IOAs represent better agreement between the simulated and observed values
(Willmott, 1982). Here, IOA was between 0.51 and 0.84 for the simulation of meteorological
parameters. Furthermore, it was not lower than 0.50 for primary air pollutants, and reached
0.81 for $O_3$ simulation at both sites. The model performances were comparable to those
reported in previous studies (Cabaraban et al., 2013; Wang et al., 2015). Therefore, we
accepted the modeling results, in view of the fact that the simulations were only used to
qualitatively indicate the interactions between the continental and marine air in this study.
Backward particle release simulations were carried out using HYSPLIT model (Stein et al.,
2015) for episode days at WS and TC sites during the entire sampling period (Draxler and
Rolph, 2003). The backward particle release simulation, which considers the dispersion
processes in the atmosphere, is capable of identifying the history of air masses (Guo et al.,
2009; Ding et al., 2013a, 2013b). In this work, we applied the model following a method
developed by Ding et al. (2013a).



**3 Results and Discussion**
**3.1 Spatio-termporal variations**
Table 1 summarizes the meteorological conditions and chemical species observed at WS and
TC. Lower temperature ($25.7 \pm 0.1\,^{\circ}$C) and higher relative humidity ($82.8 \pm 0.4$%) were
recorded at the marine site (WS) compared to the suburban site (TC) ($p < 0.01$) (temperature:
$26.7 \pm 0.1\,^{\circ}$C and relative humidity: $67.7 \pm 0.5$%). At WS, the solar radiation ($635.8 \pm 46.9$ Wm$^{-2}$)
was much higher than that at TC ($563.5 \pm 46.1$ Wm$^{-2}$, $p < 0.01$), while the average wind speed
at TC ($4.6 \pm 0.1$ms$^{-1}$) was significantly lower than that measured at WS ($7.2 \pm 0.2$ms$^{-1}$). The
lower wind speed at TC was related to the roughness of underlying surfaces. However, no
statistical differences were found for the average wind direction (about $81^{\circ}$, northeast wind)
at the two sites, indicating that the two sites were probably under the influence of similar air
masses in most cases.
The NO, NO$_2$, CO, SO$_2$ and total VOCs (the sum of NMHCs and carbonyls) had lower
average and maximum mixing ratios at WS than those at TC. The lower levels of primary air
pollutants at WS were likely the results of fewer local emission sources, faster photochemical
consumption (as discussed later) and/or more favorable dispersion conditions (*e.g.*, higher
wind speed). In contrast, O$_3$ was much higher at WS (Table 1), attributable to the
enhancements by both meteorological and photochemical effects, as discussed in sections 3.2
and 3.3.
**Table 1.** Descriptive statistics of meteorological parameters and trace gases at the two sites
during the sampling period.

| Parameter | WS | | TC | |
|---|---|---|---|---|
| | *Mean ±95% C.I.* | *Max.* | *Mean ±95% C.I.* | *Max.* |
| Temperature ($^{\circ}$C) | $25.7 \pm 0.1$ | 32.8 | $26.7 \pm 0.1$ | 35.4 |
| Relative humidity (%) | $82.8 \pm 0.4$ | 98.9 | $67.7 \pm 0.5$ | 96.8 |
| Solar radiation (W m$^{-2}$)* | $635.8 \pm 46.9$ | 1026.8 | $563.5 \pm 46.1$ | 910.0 |
| Wind speed (m s$^{-1}$) | $7.2 \pm 0.2$ | 23.8 | $4.6 \pm 0.1$ | 13.8 |
| Wind direction ($^{\circ}$) | 81.3 | - | 80.9 | - |
| O$_3$ (ppbv) | $51.3 \pm 1.2$ | 173.0 | $30.0 \pm 1.0$ | 159.9 |
| NO (ppbv) | $0.7 \pm 0.1$ | 21.0 | $14.0 \pm 0.8$ | 115.7 |





| | | | | |
|---|---|---|---|---|
| $NO_2$ (ppbv) | 4.3±0.3 | 49.3 | 25.0±0.6 | 104.2 |
| CO (ppbv) | 251.4±6.5 | 727.7 | 560.5±6.3 | 1047.9 |
| $SO_2$ (ppbv) | 2.4±0.1 | 12.2 | 5.9±0.1 | 19.1 |
| NMHCs (ppbv) | 12.7±1.1 | 32.9 | 17.7±1.7 | 60.0 |
| Carbonyls (ppbv) | 7.9±0.7 | 16.3 | 9.2±0.7 | 26.5 |

* Average of the daily maximum solar radiation. *C.I.* denotes confidence interval.
Time series of local meteorological parameters and hourly mixing ratios of air pollutants at
the two sites are illustrated in Figures 2a-2b. The temporal patterns of wind directions were
generally similar at both sites, with the dominance of the southerly winds in August and
northeastern winds between September and November. Occasionally, the northwesterly
winds from the PRD region were observed.
This sampling campaign witnessed 17 $O_3$ episodes and 7 near-$O_3$ episode days at TC, which
refers to the days with maximum hourly mixing ratio of $O_3$ higher than 100 ppbv and within
the range of 80-100 ppbv, respectively. (80 ppbv was Level I of China National Ambient Air
Quality Standard for $O_3$). At WS, 21 $O_3$ episodes and 6 near-$O_3$ episodes were recorded.
Specifically, 13 $O_3$ episode days were simultaneously observed at the two sites, with the rest
occurred exclusively at one site. On one hand, the primary air pollutants (CO, $SO_2$ and $NO_x$)
generally increased during $O_3$ episodes, implying enhanced $O_3$ formation potentials from the
precursors. On the other hand, $O_3$ episodes were always accompanied by the synoptic
conditions, *i.e.*, tropical cyclone (typhoon in the mature form) and continental anticyclone,
and/or mesoscale circulations such as SLB, as detailed in Table S4. For example, the two
multi-day $O_3$ episode events, *i.e.*, 1-8 Oct. and 19-27 Oct. (highlighted in blue in Figure 2),
were strongly associated with continental high pressure. These episode days generally had
high temperature, northerly winds, and intensive solar radiation, with air flows largely from
the inland or the coastal areas. Also, the mixing ratios of CO, $NO_2$ and $SO_2$ usually increased
during these days, suggesting the accumulation of local air pollutants and/or the increasing
contribution from regional transport. In contrast, $O_3$ episodes under the influence of tropical
cyclones (highlighted in orange in Figure 2) featured high temperature, strong solar radiation
and typically calm or moderate northwesterly to northeasterly winds, except for typhoon





"Haiyan" occurred on 9-12 Nov. (discussed in section 3.2.1). These conditions were all
conducive to the formation and accumulation of $O_3$. Additionally, SLB was also an important
factor regulating $O_3$ pollution in this region during $O_3$ episodes (Table S4). Detailed
discussions can be found in section 3.2.3.

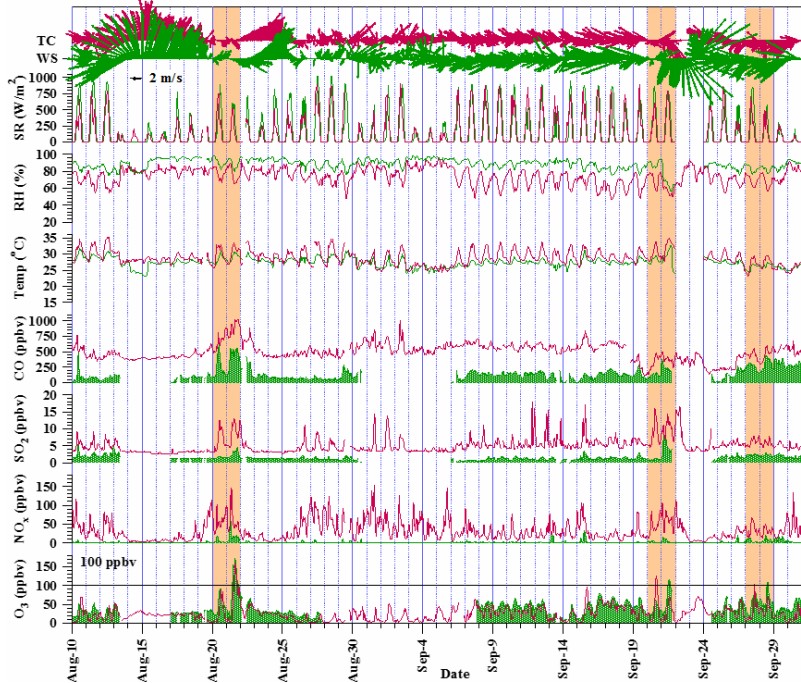

**Figure 2a.** Time series of trace gases and meteorological parameters observed for the
sampling period of 10 Aug. - 30 Sept. at WS (green) and TC (red). The black line of 100 ppbv
is the threshold for $O_3$ episode definition. The dates seriously affected by continental high
pressure and tropical cyclones are shaded in blue and orange, respectively. Note that there are
some data missing in these months due to extremely bad weather conditions and instrumental
failure.



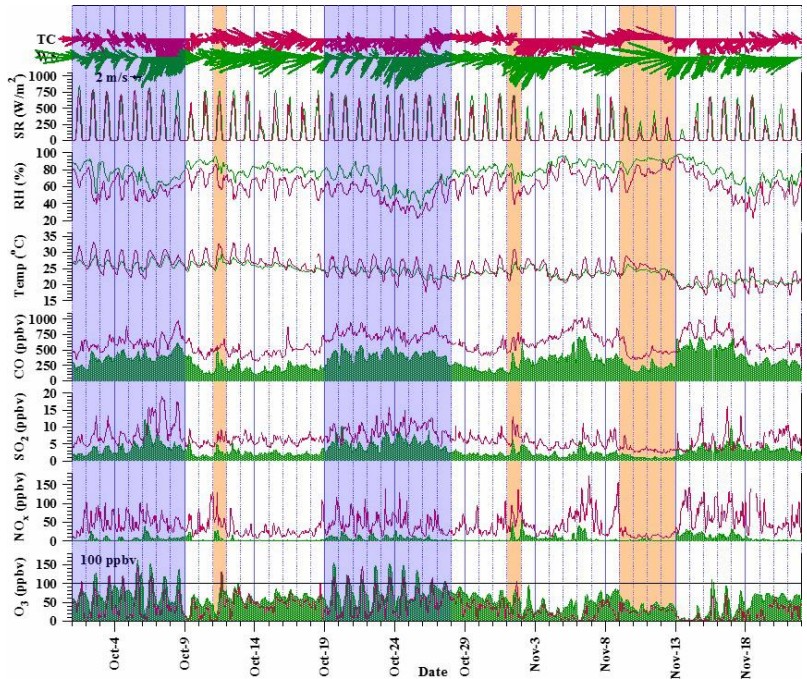

**Figure 2b.** Time series of trace gases and meteorological parameters observed for the
sampling period of 1 Oct. - 21 Nov at WS (green) and TC (red). The black line of 100 ppbv is
the threshold for $O_3$ episode definition.   The dates seriously affected by continental high
pressure and tropical cyclones are shaded in red and orange, respectively.
**3.2 Meteorological influence on $O_3$ mixing ratios**
Descriptive statistics of meteorological parameters during $O_3$ episode and non-episode days
are summarized in Table 2. On episode days the wind speed and relative humidity were lower
whereas solar radiation was stronger at both sites, suggesting that this type of weather
condition was conducive to the formation and accumulation of tropospheric $O_3$. Furthermore,
the wind direction during non-episodes was predominantly from the east (SCS), while on
episodes the winds mainly came from the north and northeast which might bring more
pollutants from the urban areas of Hong Kong and inland PRD to the sampling sites. The
characteristics of $O_3$ pollution under different weather conditions were discussed below.



**Table 2**. Descriptive statistics (Mean±95% C.I.) of meteorological parameters at the two sites
during $O_3$ episodes and non-$O_3$ episodes days.

| Parameter | WS | | TC | |
|---|---|---|---|---|
| | *$O_3$ episode* | *Non-$O_3$ episode* | *$O_3$ episode* | *Non-$O_3$ episode* |
| **Temperature ($^o$C)** | 25.3±0.2 | 25.8±0.1 | 26.3±0.3 | 26.8±0.2 |
| **Wind speed (m s$^{-1}$)** | 5.3±0.2 | 7.7±0.2 | 3.7±0.2 | 4.8±0.1 |
| **Wind direction ($^o$)** | 45.1 | 89.1 | 19.5 | 86.8 |
| **Relative humidity (%)** | 71.7±1.2 | 85.7±0.4 | 58.4±1.4 | 69.6±0.6 |
| **Solar radiation (W m$^{-2}$)*** | 723.2±26.1 | 613.7±57.6 | 699.0±29.1 | 537.0±53.1 |

* Average of the daily maximum solar radiation. *C.I.* denotes confidence interval.
*3.2.1 Tropical cyclones*
Tropical cyclone (low-pressure system) is one of the main meteorological conditions
conducive to the occurrence of $O_3$ episodes in Hong Kong (Yin, 2004; Ling et al., 2013). In
this study, 7 episode days and 3 near-episode days were closely associated with 5 tropical
cyclones (*i.e.*, Trami, Usagi, Wutip, Nari and Krosa) (Table S4 and Figure S2). For example,
Trami caused the worst $O_3$ episode on 21 Aug. with the highest peak hourly $O_3$ mixing ratios
of 160 and 173 ppbv at TC and WS, respectively. These episode or near-episode days usually
appeared 1-2 days before the arrival of the tropical cyclones, because the large-scale
peripheral subsidence of the tropical cyclones usually creates the meteorological conditions
favorable to the formation and accumulation of $O_3$, such as inversion layer, high temperature,
low humidity, intensive light, and weak winds (Wang et al., 1998; Yin, 2004). The tropical
cyclones also cause anti-clockwise air flows at their outskirt affecting the wind directions and
subsequent the regional transport of air pollution. Figure 3 illustrates surface wind fields and
air movement two days (*i.e.*, 20-21 Sept.) before the occurrence of Usagi as an example. It
can be seen that when Usagi approached southeastern area of Hong Kong, it led to weak
northeasterly and later northwesterly winds which potentially delivered $O_3$ and its precursors
from highly polluted inland PRD region to the sampling sites (Yin, 2004; Wei et al., 2016;
Wang et al., 2017a). The wind speed was lower than 4 m s$^{-1}$ at the sampling sites and in their
surrounding area on 20 Sept. (Figure 3a), and it gradually increased on the next day (21 Sept.)



with the approaching of the tropical cyclone (Figure 3b). It is noteworthy that the rarely
occurred westerly and northwesterly winds caused tropical cyclones resulted in
unsynchronized occurrence of $O_3$ episodes between the two sites (Figures 3c & d). Namely,
high $O_3$ values were observed at TC only on 20 Sept., while $O_3$ started to increase at WS on
the next day (21 Sept.). This discrepancy might indicate the transport of $O_3$ and/or its
precursors from terrestrial area to the offshore site driven by tropical cyclone.
Please note, not all tropical cyclones would cause high levels of $O_3$. For example, the tropical
cyclone Haiyan observed on 9-12 Nov. over the SCS did not cause high $O_3$ levels (Figure 2b).
Because the origin of Haiyan was at a lower latitude (southern Guam) and it moved on the
waters southwest of PRD (Figure S2), the anti-clockwise air flow caused easterly and
southeasterly winds in the north and northeast outer band of Haiyan. The winds originated
from SCS brought in clean marine air to the sampling sites, resulting in dilution and
dispersion of local air pollutants.
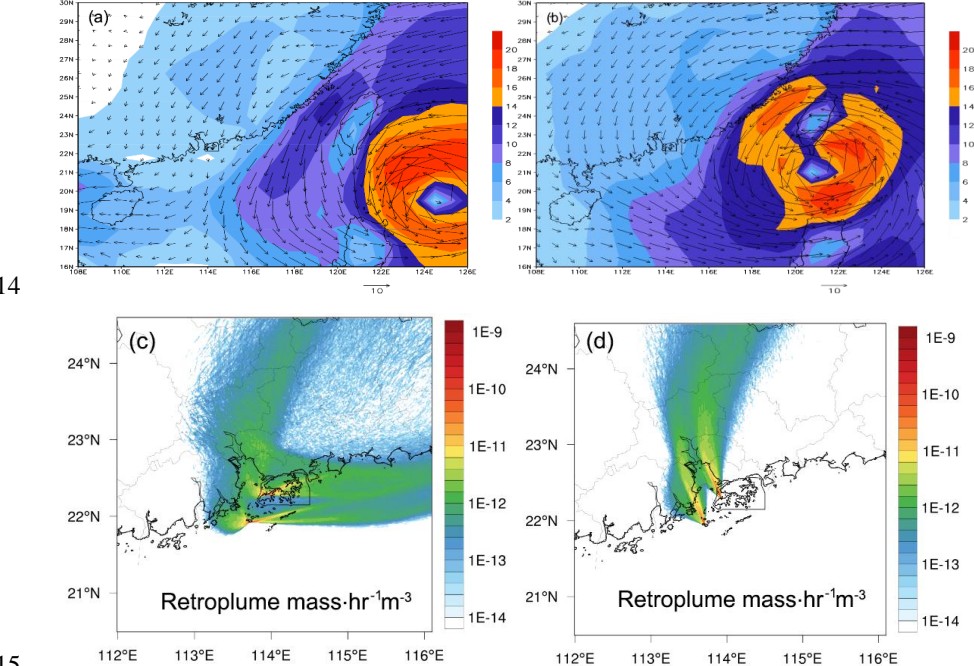
**Figure 3.** Model simulated 10 m wind vectors (arrows) and wind speed (shaded, unit: m s$^{-1}$),
and the distribution of air mass concentrations (unit: mass hr$^{-1}$ m$^{-3}$) within surface 100 m
simulated by HYSPLIT Lagrangian backward particle release model with WS and TC as the





starting points two days (20 Sept. 2013) before (a, c) and one day (21 Sept. 2013) before (b, d)
the arrival of Usagi.
*3.2.2 Continental anticyclones*
In addition to tropical cyclones, the continental anticyclone (high-pressure system) was
frequently observed in the region, which often caused high $O_3$ concentrations. For example,
two multi-day $O_3$ episodes (1-8 Oct. and 19-27 Oct.) occurred at the sampling sites when
there were intensive continental anticyclones and weak Western Pacific Subtropical High
(WPSH) to the north of Hong Kong (see Figure S3 as examples).
The main feature of the anticyclones is sinking air at the center with gentle clockwise winds
in the northern hemisphere. The air warms up as it sinks by compression leading to warm,
cloudless and dry weather, which is conducive to intensive photochemical $O_3$ formation. In
addition, anticyclone is a large-scale weather system which produces long-lasting settled and
calm weather for many days or weeks favorable to the accumulation of primary and
secondary pollutants.
Indeed, the two continental high pressure systems observed in this campaign lasted 8 and 9
days, respectively, with the presence of SLBs occasionally (*i.e.*, 2-5 Oct. and 19-21 Oct.) on
the first several days when the synoptic winds were relatively weak. The clockwise and slow
movement of the air masses caused northeasterly and easterly winds to the sampling sites and
brought in densely polluted air from the inland (Figure 2b) to the coastal areas of the SCS.
For example, the CO mixing ratios were significantly elevated during these episode days,
with an average of 409 and 683 ppbv at WS and TC, respectively, which were higher than
other episode days. The continuous input of exotic air pollutants provided essential "fuel" to
local photochemical production of $O_3$, leading to the severe multi-day $O_3$ episodes.
*3.2.3 Sea-land breeze (SLB) circulation*
During the sampling period, SLB circulations in the study area were identified on 21 out of
104 sampling days. The occurrence frequency was comparable to that reported by Zhang and
Zhang (1997) who discovered 70 SLB days in a year in the same region. In this study, 12 $O_3$
episode days were thought to be influenced by SLB (see Table S4), with 5 of them (27-28
Sept., 11-12 Oct. and 1 Nov.) under the dominance of tropical cyclones (*i.e.*, Wutip, Nari and
Krosa) and the other 7 days in association with the continental anticyclones. In addition to the



effects of tropical cyclones and continental anticyclones discussed above, SLB also posed
non-negligible impact on $O_3$ pollution in these cases.
SLB circulation is driven by sea-land thermal difference and topographic conditions, and
usually happens when the synoptic winds are weak (Liu et al., 2002; Lo et al., 2006; Lu et al.,
2009b). In general, the temperature difference between the sea and the land is large on the
SLB days. Taking 3 Oct. as an example, the maximum hourly temperature at TC was 3.2 $^{\circ}$C
higher than that at WS during daytime hours, whereas the minimum hourly temperature in the
evening was 2.7 $^{\circ}$C lower at TC than at WS. On a typical SLB day, wind blows onshore
during the day (sea breeze) and offshore in the evening (land breeze). However, the transition
time of breezes in this study was found to vary in a wide range. The sea breeze switched to
land breeze between 00:00 and 08:00 with a median of 03:00 for breeze shifting, and 11:00 –
18:00 with the median of 14:00 was the time when land breeze turned to sea breeze. Ding et
al. (2004) also reported this phenomenon and pointed out that the start time of sea breezes in
Hong Kong was generally delayed to noontime due to the synoptic northerly winds blowing
from the continental areas to SCS, particularly on $O_3$ episode days when northerly winds
dominated in Hong Kong. For example, the sea breeze commenced at 15:00 on 3 Oct. and
transited to land breeze at 4:00 on 4 Oct. (Figure 4). Figures 4a and 4b depict the surface
wind fields with a sea breeze and a land breeze, respectively. The vertical wind fields with
the sea breeze and land breeze are presented in Figures 4c and 4d, respectively. Surface and
vertical SLB circulations were clearly seen in these panels of Figure 4. The mesoscale
circulations caused by SLB might promote the interactions between the continental (TC) and
marine (WS) atmospheres. Specifically, the primary air pollutants observed at TC could be
transported to WS by land breeze. Moreover, the air masses could return to TC after
sufficient photochemical evolutions over SCS, during which $O_3$ might also be elevated in the
continental areas.



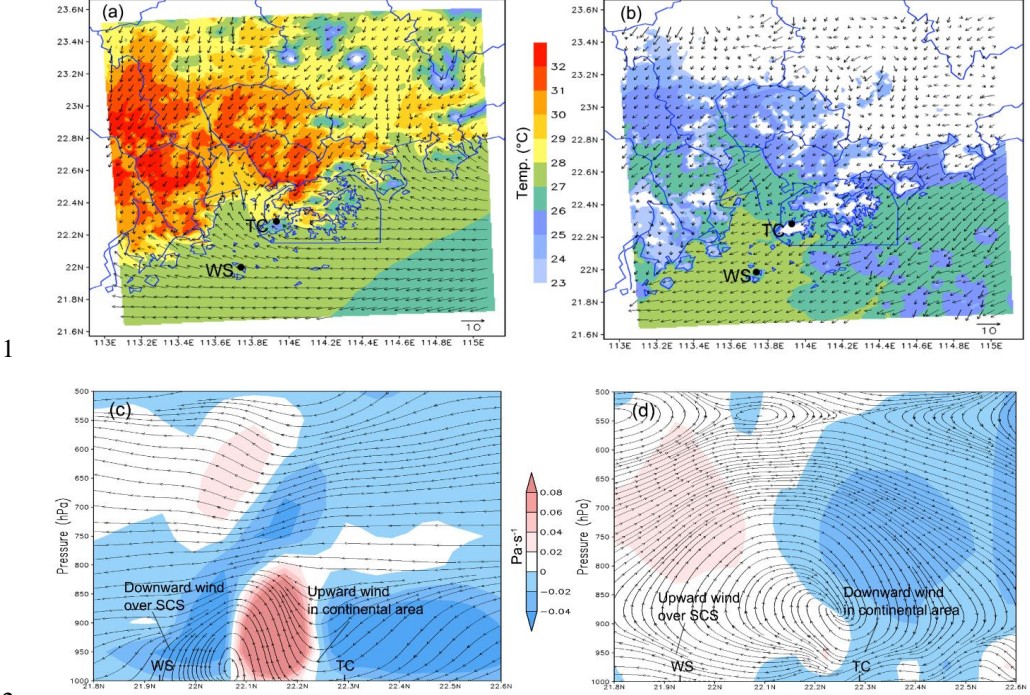

**Figure 4.** SLB circulation on 3-4 Oct. 2013, showing surface wind pattern (arrows) and
temperature (color) at 17:00 on 3 Oct. (a) and at 04:00 on 4 Oct. (b). Vertical cross-section
(taken over a longitude of 113.85 °E, mean of the longitudes of TC and WS) depicting the v–
w wind stream (arrow) and the index of $\omega$ *100 (color) at 17:00 on 3 Oct. (c); and at 04:00 on
4 Oct. (d). For figures (c) and (d), the blue color (negative) and light red color (positive)
present downward and upward winds, respectively. Figures (a) and (c) represent a sea breeze,
and Figures (b) and (d) show a land breeze. Note that $\omega$ is the vertical velocity in isobaric
coordinates.
**3.3 Chemical characteristics of air masses**
*3.3.1 Chemical composition*
To inspect the chemical characteristics of air masses on $O_3$ episode days and non-$O_3$ episode
days, chemical species are statistically summarized at the two sites (Table 3). As expected,
the levels of all pollutants (*i.e.*, $O_3$, $NO_2$, CO, $SO_2$, NMHCs and carbonyls) were significantly
higher on $O_3$ episode days for both sites ($p<0.05$), except for the comparable or even lower





NO due to its titration to $O_3$ (see Section 3.3.2). Table S5 shows the top 10 NMHC species
observed during $O_3$ episodes and non-episodes at the two sites. The dominant species were
quite similar regardless of episode or non-episode days at both sites. The higher
concentrations of both primary and secondary pollutants on episode days than those on non-
episode days were likely due to more intense photochemical reactions, more local pollutant
accumulation as well as the regional transport of more highly polluted air masses. On the
other hand, the similar NMHCs composition at both sites during both episodes and non-
episodes indicated somewhat interaction of air masses between the two sites regardless of $O_3$
levels.
It is worth to mention that $O_3$ was much higher at WS than that at TC during both episodes
and non-episodes ($p<0.01$), with an average difference of 30.2 ppbv and 16.7 ppbv,
respectively (Table 3), though the levels of $O_3$ precursors (*i.e.,* $NO_x$ and VOCs) at WS were
lower. Insight into VOC ratios found that ethene/ethane (0.5±0.04) and toluene/benzene
(2.2±0.5) at WS were significantly ($p<0.05$) lower than those at TC (0.7±0.1 and 2.9±0.4,
respectively), likely indicating that the air masses at WS were more aged (Guo et al., 2007).
Therefore, the higher $O_3$ at WS might be partially attributable to the aging of air masses (*e.g.*,
during the transport of continental air).
**Table 3**. Descriptive statistics (Mean±95% *C.I.*) of measured air pollutants, simulated OH
and $O_3$ production rates at the two sites during $O_3$ episodes and non-$O_3$ episodes days.

| Parameter | WS | | TC | |
|---|---|---|---|---|
| | *$O_3$ episode* | *Non-$O_3$ episode* | *$O_3$ episode* | *Non-$O_3$ episode* |
| $O_3$ (ppbv) | 74.3±3.0 | 43.9±1.0 | 44.1±3.6 | 27.2±0.8 |
| $O_x$ (ppbv) | 81.6±2.9 | 47.8±1.0 | 83.3±3.7 | 49.4±1.0 |
| NO (ppbv) | 0.6±0.1 | 0.7±0.1 | 11.5±1.4 | 14.5±0.9 |
| $NO_2$ (ppbv) | 7.3±0.6 | 3.3±0.3 | 39.2±1.7 | 22.2±0.6 |
| CO (ppbv) | 391.4±9.1 | 209.4±6.8 | 652.9±16.0 | 541.9±6.5 |
| $SO_2$ (ppbv) | 4.3±0.2 | 1.9±0.1 | 8.1±0.3 | 5.5±0.1 |
| NMHCs (ppbv) | 17.7±1.4 | 9.6±1.2 | 20.2±2.2 | 16.8±2.1 |
| Carbonyls (ppbv) | 10.3±0.8 | 5.4±0.4 | 12.0±1.3 | 8.1±0.7 |
| $NO_2/NO$ (ppbv/ppbv) | 12.7±1.1 | 4.7±0.5 | 3.4±0.4 | 1.5±0.2 |
| Simulated OH ($\times 10^6$ molecules $cm^{-3}$) | 5.4 ±1.0 | 3.3 ±0.8 | 1.2 ±0.3 | 1.5 ±0.3 |



* Average of the daily maximum solar radiation. *C.I.* denotes confidence interval. $O_x = O_3 + NO_2$.
*3.3.2 Influence of NO titration*
Apart from the age of air masses, NO titration is another important factor influencing $O_3$
concentration. In areas with high NO levels, the NO titration ($O_3 + NO \rightarrow NO_2 + O_2$) is a
main process consuming $O_3$. In this study, the average NO mixing ratio at TC was $14.0\pm0.8$
ppbv, compared to $0.7\pm0.1$ ppbv at WS (Table 1). The much lower NO at WS implied
weaker titration to $O_3$, which enabled the survival of $O_3$ in high concentration. A direct
evidence of NO titration effect was the trough of $O_3$ during the morning rush hours (06:00-
07:00), together with an increase of $NO_2$ (Figure S4). Furthermore, the total oxidants ($O_x =$
$O_3 + NO_2$), which are usually adopted to take into account the NO titration influence, were
comparable ($p > 0.05$) between TC and WS with mean values of $83.3\pm3.7$ ppbv and $81.6\pm2.9$
ppbv during $O_3$ episodes, and $49.4\pm1.0$ ppbv and $47.8\pm1.0$ ppbv during non-episodes,
respectively (Table 3). This was reasonable in view of the interactions between the two sites.
However, the remarkably higher $O_3$ and lower NO at WS indicated that NO titration was a
determinant factor regulating the $O_3$ levels at both sites.
Moreover, NO titration is generally more significant on high $O_3$ days, resulting in higher
$NO_2/NO$ ratios due to the conversion of NO to $NO_2$ by $O_3$. Indeed, the mean $NO_2/NO$ ratios
increased from $4.7\pm0.5$ at WS and $1.5\pm0.2$ at TC during non-episodes to $12.7\pm1.1$ and
$3.4\pm0.4$ during $O_3$ episodes, respectively, implying that more $O_3$ was titrated by NO during
episodes. As a result, NO at TC was lower ($p < 0.01$) during $O_3$ episodes than during non-
episodes (Table 3). It is noteworthy that NO at WS was on the same level between $O_3$ episode
and non-$O_3$ episode days ($p > 0.05$). This probably related to the weak titration at this marine
site due to the trivial NO concentrations in both periods, as well as the counteracting effect of
the increased transport of NO under northerly winds against the enhanced titration during $O_3$
episodes.
The aforementioned discussion demonstrated that NO titration played an important role in
altering $O_3$ distribution, especially on $O_3$ episodes days. The lower NO (weaker NO titration)
partially resulted in the higher $O_3$ concentrations observed at WS.



*3.3.3 Atmospheric oxidative capacity*
The $O_3$ formation is generally initiated by the oxidations of VOCs by OH. Furthermore, the
oxidative radicals (*e.g.*, $RO_2$) generated from these reactions experience an array of
transformations, through which OH can be recycled. Thus, the OH concentration is an
important indicator to evaluate the atmospheric oxidative capacity and the potential of $O_3$
formation. As shown in Table 3, the OH concentration simulated by PBM-MCM model was
significantly higher ($p<0.05$) at WS than that at TC, regardless of $O_3$ episode ($5.4 \pm 1.0 \times 10^6$
molecules $cm^{-3}$) or non-episode days ($3.3 \pm 0.8 \times 10^6$ molecules $cm^{-3}$). This indicated that the
oxidative capacity of the atmosphere at WS was stronger than that at TC, which might
explain the higher $O_3$ at WS. Moreover, while the simulated OH remained unchanged
($p>0.05$) at TC, it increased largely ($p<0.05$) from non-episodes to episodes at WS,
suggesting that the oxidative capacity of the atmosphere at WS was more enhanced during $O_3$
episodes.
Furthermore, we simulated the rates of $O_3$ production, destruction and net $O_3$ production at
both sites, as presented in Figure 5. No significant change in net $O_3$ production was observed
between $O_3$ episode ($1.1\pm0.6\times10^7$ molecules $cm^{-3}$ $s^{-1}$) and non-episode days ($1.2\pm0.3\times10^7$
molecules $cm^{-3}$ $s^{-1}$) at TC ($p>0.05$). Since previous studies (Cheng et al., 2010a; Wang et al.,
2017a) repeatedly confirmed that $O_3$ formation at TC was limited by VOCs, the unchanged
net $O_3$ production might be due to the balance between the increased $O_3$ production and $O_3$
destruction resulting from the elevated VOCs and $NO_x$ during $O_3$ episodes, respectively. On
the contrary, the net $O_3$ production increased remarkably from non-episodes ($1.2\pm0.2\times10^7$
molecules $cm^{-3}$ $s^{-1}$) to $O_3$ episodes ($3.9\pm0.8\times10^7$ molecules $cm^{-3}$ $s^{-1}$) at WS. Insight into the $O_3$
formation and destruction pathways found that the slight increases of $O_3$ destructions (mainly
through $OH+NO_2$ and $O^1D+H_2O$) were overridden by the great enhancements of $O_3$
productions through $RO_2+NO$ and $HO_2+NO$. Our recent study (Wang et al., 2017b) revealed
that $O_3$ formation at WS was in a transition regime and much more sensitive to $NO_x$ during
non-episodes, when the principal photochemical reaction pathways to produce $O_3$ (*i.e.*,
$RO_2+NO$ and $HO_2+NO$) were seriously limited by the low $NO_x$ levels. During $O_3$ episodes,
with the increase of $NO_x$ (Table 3), the contributions of the aforementioned two $O_3$
production pathways were significantly enhanced (Figure 5). In addition, the increased VOCs
also contributed to the $O_3$ formation by producing more $RO_2$ radicals (not shown here)
through OH-initiated reactions. Therefore, the combined effect of elevated VOCs and $NO_x$





during $O_3$ episodes at WS was the increase of $O_3$ production, which was insignificant at TC.
Detailed discussion on the $O_3$ photochemistry at WS can be found in our recent publication
(Wang et al., 2017b).

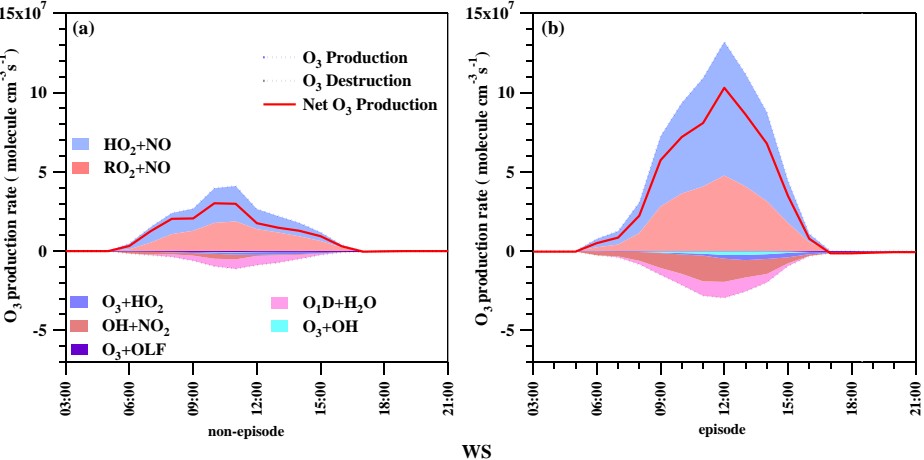

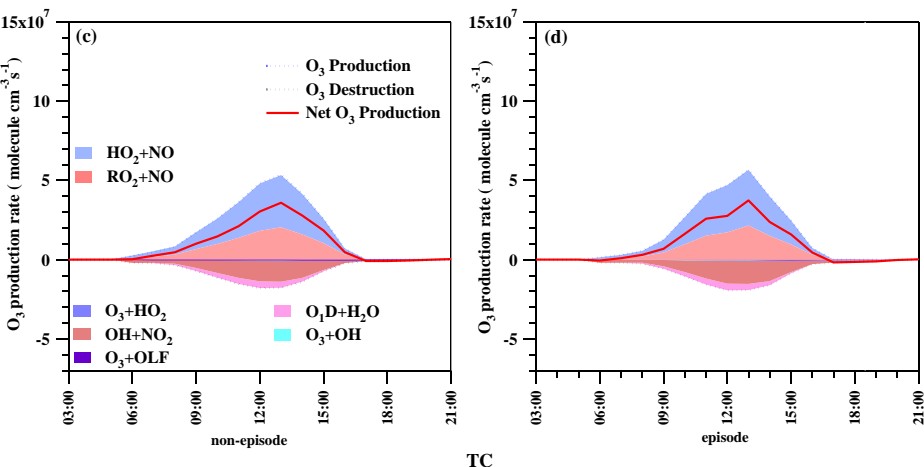

**Figure 5**. Simulated rates of $O_3$ production, destruction, and net $O_3$ production (unit:
molecules $cm^{-3}$ $s^{-1}$) on $O_3$ episode and non-episode days at WS (panels (a) and (b)) and TC
(panels (c) and (d)). Panels (a) and (b) were adapted from our recent publication (Wang et al.
2017b).



### 3.4 Impact of air mass interaction on $O_3$ pollution in coastal areas

Driven by various weather systems (*e.g.,* continental anticyclones, WPSH, tropical cyclones and SLBs), continental and marine air masses frequently interact with each other in the coastal areas. When continental air masses intrude into marine atmosphere, the chemical composition and atmospheric oxidative capacity over the marine atmosphere will be altered by the introduction of anthropogenic pollutants. Taken 21 Aug. as an example, when the sampling sites (TC and WS) were under northwesterly to southwesterly winds caused by tropical cyclone (Figure 2a), the maximum hourly $O_3$ reached 160 and 173 ppbv at TC and WS, respectively. Correspondingly, the primary air pollutants all stayed on high levels, compared to those during non-episodes (Figure 2a). Since WS was almost free of anthropogenic emissions, the great abundances of both primary and secondary air pollutants implied the influence of continental pollution on air quality at this site. Figures 6-7 depict the spatial distributions of CO and $O_3$ over the region of interest at selective time (08:00, 14:00, 19:00 and 23:00) on 21 Aug., respectively. CO is presented as an example of primary air pollutants emitted from anthropogenic sources. The spatiotemporal patterns of CO and $O_3$ were simulated by WRF-CMAQ. Noticeably, the model well reproduced high level of CO in PRD region at 08:00, which was reasonable in view of the vehicular emissions in urban areas during morning rush hours. However, under the dominance of northwesterly winds in the morning, the center of high CO moved to the coastal areas. Even though the winds changed to southwesterly at noon, CO concentration over SCS was still remarkably elevated according to the simulated results at 14:00. Further, the spatial distribution of CO at 19:00 and 23:00 confirmed the continuous movement of the polluted air masses away from the land under southwesterly winds. It should be noted that the increase of CO in PRD region at 19:00 and 23:00 were most likely caused by the vehicle emissions during evening rush hours. Overall, the dynamic distribution of CO in the study area clearly indicated the interaction between continental and marine atmospheres. As a result of the intrusion of continental air, high level of $O_3$ was simulated over SCS at 14:00 (Figure 7b), which was comparable to the observed value (148 ppbv) at WS. Moreover, $O_3$ was even higher over SCS than that in continental area, due mainly to the more aged air masses, lower NO titration and higher oxidative capacity of the atmosphere (see section 3.3). Consistent with CO, the center of high $O_3$ moved away from the land. At 19:00, the $O_3$-laden air mass penetrated into the SCS ~300 km, causing ~8,000 $km^2$ water area (8 times the area of Hong Kong) under high level of $O_3$ (>100





ppbv). This case provided solid evidence of the transport of continental air masses to SCS,
which aggravated air pollution (particularly $O_3$ pollution) in this offshore area.

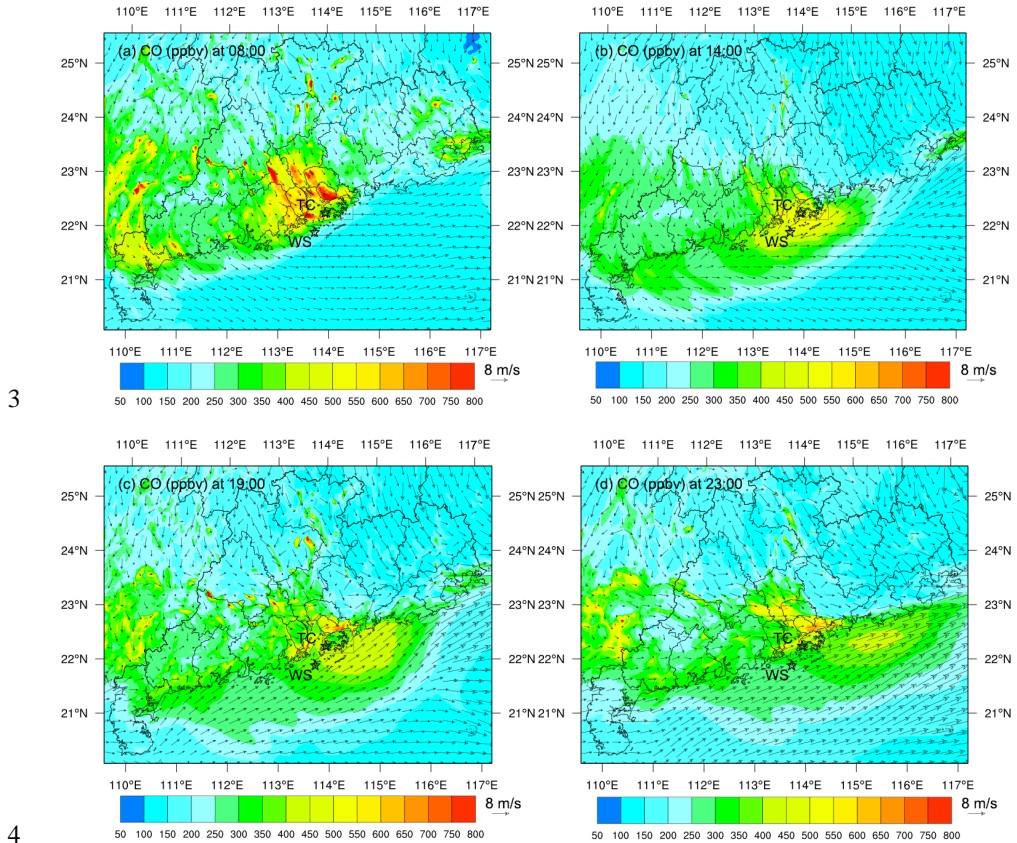

**Figure 6.** Spatial distribution of CO at 08:00 (a), 14:00 (b), 19:00 (c) and 23:00 (d) on 21
August simulated by WRF-CMAQ, taken as an example of the "Outflow" interaction pattern.
Arrows in the figure represent the surface wind field.



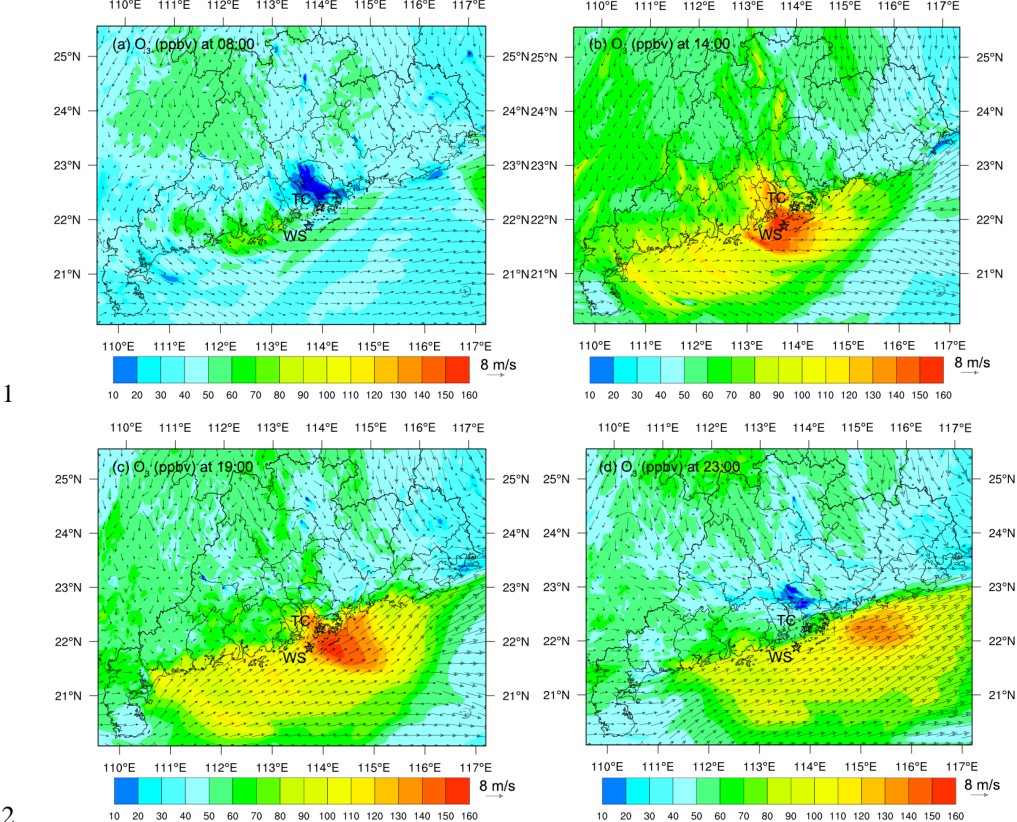

**Figure 7** Spatial distribution of $O_3$ at 08:00 (a), 14:00 (b), 19:00 (c) and 23:00 (d) on 21 August simulated by WRF-CMAQ, taken as an example of the "Outflow" interaction pattern. Arrows in the figure represent the surface wind field.

In contrast to outflow of continental air masses, the continental area near the coast could also be immersed by oceanic air under sea breeze. Contrary to our general expectation that ocean airflow dilutes air pollution, the sea breeze carrying elevated $O_3$ formed over SCS might build up the terrestrial $O_3$ in the coastal area in some cases. Figure 8 shows the spatial distribution of $O_3$ over the study area on 3 Oct., as an example of SLB regulating $O_3$ formation and distribution (see Figure 4). Similar to the aforementioned scenario controlled by tropical cyclone, the simulated $O_3$ at 14:00 was generally higher over SCS than in the terrestrial area, indicating the transport of polluted air masses from the land to the sea area. This was confirmed by the prevailing northeasterly winds in the morning (08:00 here). However, the $O_3$-laden air did not move far away from the land subsequently. Instead, it progressively approached the land, leading to increase of $O_3$ concentration in most parts of




Hong Kong. This is because the wind direction in the coastal region changed from
northeasterly to southeasterly at 17:00. Namely, the sea breeze appeared in late afternoon,
which delivered the high $O_3$ formed over SCS to the continental areas near the coast. In fact,
the air quality monitoring stations deployed in southern Hong Kong by HKEPD also recorded
the $O_3$ peak in the evening when $O_3$ could not be formed locally (Figure S5), further
confirming the recirculation of $O_3$-laden air from SCS to coastal areas of Hong Kong under
sea-breeze. However, the oceanic air did not penetrate further into the inland PRD, which
was likely stopped by the strong northeasterly winds dominated in the inland areas. Overall, it
can be seen that SLB as a common interaction between marine and continental atmospheres
played important role in regulating $O_3$ formation and distribution in coastal region of SCS,
which is also applicable to other similar mesoscale environments over the world.

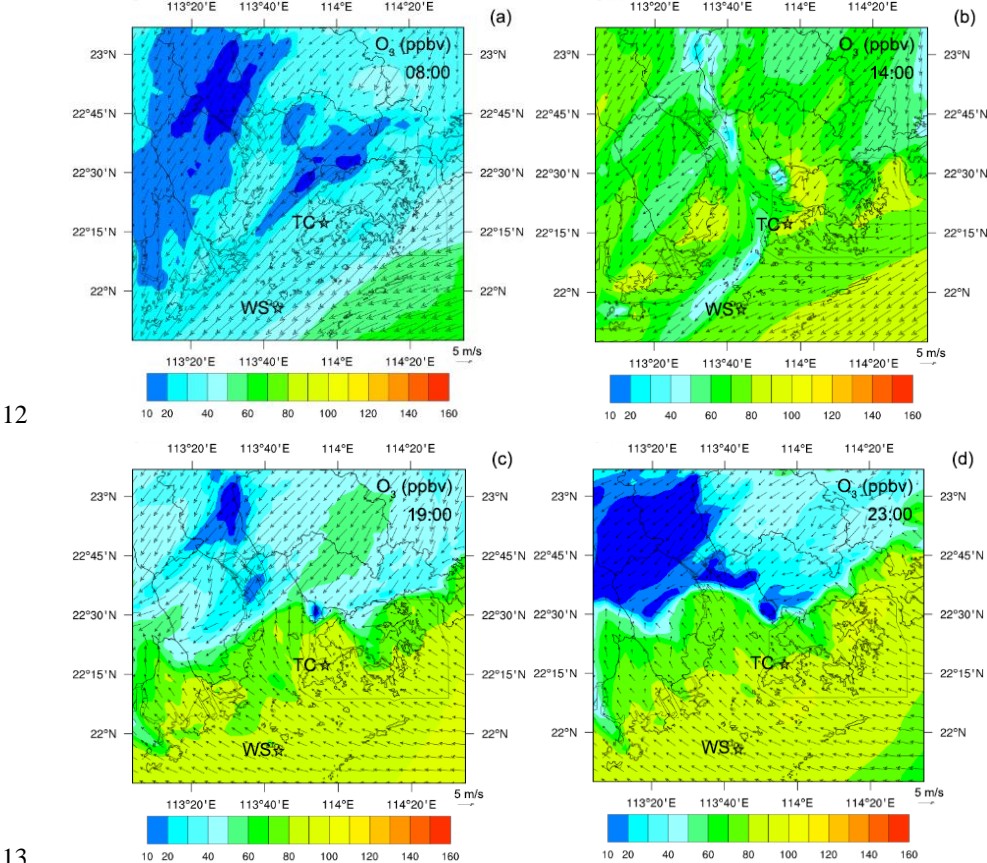



**Figure 8.** Spatial distribution of $O_3$ at 08:00 (a), 14:00 (b), 19:00 (c) and 23:00 (d) on 3 October, taken as an example of the "SLB" interaction pattern. Arrows in the figure represent the surface wind field.

## 4. Conclusions

Coastal regions with dense population, economic prosperity and environmental pollution are common in the world. This study provided an overview of $O_3$ pollution in warm seasons around a coastal region of SCS, focusing on the influences of interactions between marine and continental atmospheres on air quality in this subtropical region. The concurrent measurements of primary and secondary air pollutants at TC (a continental site) and WS (a marine site) from August to November 2013 indicated that $O_3$ was much higher at WS than that at TC, contrary to the more abundant primary air pollutants at TC. At the two sites, $O_3$ episodes and near-$O_3$ episodes were frequently observed, which were closely associated with continental anticyclone, tropical cyclone and SLB. In addition to high temperature, strong solar radiation and weak wind, the aforementioned meteorological conditions all favored the transport of polluted air masses from continental areas to SCS, during which the air pollutants were transformed with the aging of air masses. After arriving in SCS, the land-originated air pollutants further involved in intensive photochemical reactions with the trait of low NO titration to $O_3$ and high $O_3$ production rate, leading to higher $O_3$ level in marine atmosphere (WS) than that in coastal cities (TC). In addition to the continental outflow that aggravated $O_3$ pollution over SCS, SLB as a common interaction in coastal areas also often facilitated the recirculation of $O_3$ formed over SCS to the continental areas, building up $O_3$ concentration in coastal cities under sea breeze. The findings can be extended to other similar regions to advance our understanding of $O_3$ pollution.

**Acknowledgements**

This project was supported by the Natural Science Foundation of China (Grant No. 41275122), the Research Grants Council (RGC) of the Hong Kong Government of Special Administrative Region (PolyU5154/13E, PolyU152052/14E, PolyU152052/16E and CRF/C5004-15E), the Guangdong special fund for science and technology development (2017B020216007), and partly by the Hong Kong PolyU internal grant (G-SB63, 1-BBW4 and 4-ZZFW). The authors thank HKEPD for provision of the air quality and meteorological data at TC site, and are grateful to Po On Commercial Association Wan Ho Kan Primary





School at Tung Chung and the National Marine Environmental Monitoring Station at
Wanshan Island for their generous support on the field study. Contributions to field
measurements by Kalam Cheung, Dawei Wang, Bo Liu, Nan Wang, Jiamin Ou, Huanghuang
Yan and Xiaoxin Fu are also highly appreciated. The authors also gratefully acknowledge the
NOAA Air Resources Laboratory (ARL) for the provision of the HYSPLIT transport and
dispersion model and/or READY website (http://www.ready.noaa.gov) used in this
publication.

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
