# Peer review of "Ozone pollution around a coastal region of South China Sea: Interaction between marine and continental air"

_Atmospheric Chemistry and Physics, 2017_

## Referee Comment (RC1) · Anonymous Referee #1 · 15 Jan 2018

This paper provides a detailed study of trace gases and meteorology at two sites, one a suburban site in Hong Kong (labelled TC), the other a coastal site (labelled WS), with few local anthropogenic emissions, on the edge of the South China Sea. The sites are separated by $\sim$ 40 km. Emphasis is given to ozone episodes (>100 ppbv) and near episodes, which occurred on a number of occasions, some extending over 9 days, during two $\sim$50 day periods in August, September; October, November 2013. The results are rationalised in detail using a range of modelling techniques: a zero dimensional box model study using the master chemical mechanism (MCM); the Weather Research and Forecasting (WRF) model to provide wind fields and coupled with the CMAQ model to provide an Eulerian representation of the physical and chemical processes over a wide

area and HYSPLIT to provide backward particle release simulations to understand air mass origins. The paper provides a very useful dataset and an interesting analysis. The results are discussed in terms of the interaction between marine and continental air. The episodic ozone concentrations are significantly higher at WS than at TC and much of the paper relates to a discussion of the origin of these differences, which are ascribed to weaker NO titration and to a stronger oxidative capacity at the coastal site. The main meteorological features during the episodes were tropical cyclones, with transport from the polluted Pearl River Delta Region to the sites, continental anti-cyclones, which again brought air from polluted inland areas and Sea Land Breezes, with alternation of onshore and offshore winds. My main concern is with the contention that the results relate to the interaction between marine and continental air, which is included in the title and pervades the text. WS is one of several islands lying close to the coast. Its important characteristic is that there are few local emissions so that NOx is low. Other pollutants, CO, SO2, NMHC show clear indications of advection of polluted air, but the concentrations are on average lower than those found at TC. The wind patterns confirm that the air is primarily, perhaps exclusively during the episodes, of continental origin. Even the SLB winds from the sea simply advect high ozone concentrations, formed in polluted air, back to the coastal region. Marine air has much less impact than is found and has been widely discussed at, say, Mace Head in Ireland or Cape Grim in Tasmania. The observation of higher ozone at WS compared with TC derives primarily, as argued, from the low emissions at WS and the consequently much lower NOx and reduced titration via NO + O3. The most telling observation is the near equivalence of the total oxidant concentration at the two sites during both episodes and non-episodes (p 20). Similar behaviour is of course found in many other locations when comparing rural and urban ozone concentrations in similar air masses. It is the absence of local NOx emissions at WS that leads to the differences; it is not specifically related to its coastal location and certainly not to marine influences. The discussion of the daily ozone profile could also be improved. • The diurnal variation is superimposed on a residual night-time ozone concentration, which is substantial,

[Figure]

Figs 2 and S4. This might be discussed. • Is the higher rate of ozone formation, shown in Fig 5b, a reflection of the high ozone concentration itself? It would be helpful to show the concentrations of OH, HO2 and RO2 vs time and also their rates of production and loss. Is the enhanced [OH] a result of increased O1D production from the higher [O3] found at WS in episodes? These plots could, if necessary, be shown in the Supplement. • It would also be helpful, again in the Supplement, to see ozone, OH, HO2 and RO2 concentrations, and ozone and radical rates of formation and loss on a specific episode day. Using averages can lead to a loss of clarity and understanding. Two additional points: O1D in the caption to Fig 5 should be O1D. The English needs a good deal of attention, particularly the frequent absence of definite / indefinite articles. The paper makes a substantial contribution and should be published in ACP. The authors, though, should consider the points made above relating to the overall emphasis of the paper and the clarity of the discussion on chemical processes.

(report also included as pdf)

Please also note the supplement to this comment:
https://www.atmos-chem-phys-discuss.net/acp-2017-988/acp-2017-988-RC1-supplement.pdf
* * *

---

## Referee Comment (RC2) · Anonymous Referee #2 · 29 Jan 2018

This paper reports intensive field measurements at two sites over the South China Sea. The spatial distribution of ozone pollution and its favorable synoptic conditions were interpreted. The authors also tried to link, by the sea-land breeze, the transport of continental pollution to oceans and the recirculation of land-originated aged air masses from ocean to the coastal regions. The manuscript is generally well written and easy to follow. The following specific comments should be addressed before it can be considered for publication at ACP.

Specific Comments:

Page 1, Line 6: Hong Kong, China

[Figure]

Page 2, Lines 5-6: the authors may either spell out TC and WS, or just remove them from the abstract.

Page 2, Line 15: the word "magnified" may be not appropriate here. The ozone-laden air may be transported to a larger area over the oceans, but should not be "magnified".

Page 3, Lines 14-19: to date, the long-term O3 trend studies were relatively limited in China. The authors should refer to the following earlier studies in Hong Kong, the PRD region and northern China.

Sun, et al., Significant increase of summertime ozone at Mount Tai in Central Eastern China, Atmos. Chem. Phys., 16, 10637-10650, 2016.

Xue et al., Increasing external effects negate local efforts to control ozone air pollution: a case study of Hong Kong and implications for other Chinese cities, Environ. Sci. Tech., 48, 10769-10775, 2014.

Page 4, Lines 24-26: a more recent study has investigated the detailed chemical features including the radical chemistry in different air masses arriving at the South China Sea.

Li et al., Oxidizing capacity of the rural atmosphere in Hong Kong, Southern China. Science of the Total Environment. 612. 1114-1122. 2018.

Page 5, Line 21-22: it is not the case that northeast monsoon prevails in late summer. The O3 episode occurring in late summer in Hong Kong is mainly related to the tropical cyclones.

Page 6, Lines 8-10: it has been known that the traditional commercial NOx analyzer may be subject to significant positive interference for the NO2 measurements, especially at the rural and remote areas like WS. The authors need state the uncertainty of the NO2 measurements and the subsequent observation-based modeling analysis.

Page 8, Lines 16-18: so the OBM was not constrained by the measured HONO and

OVOCs, right? This may affect the accurate modeling of OH radicals and ozone formation. As the OVOC measurements were available in the present study, the authors should constrain the model with the measured OVOC data.

Page 9, Lines 19-27: it would be better if the authors could provide the time series of model simulations and observations for a direct comparison, maybe in the supporting information.

Page 10, Table 1: it would be much better if the statistics of the most abundant NHMC and carbonyl species are individually shown, instead of the bulk concentrations.

Section 3.2: this section is too long and contains a lot of general description of the typhoon, continental anticyclone, and sea-land breeze (most of them are already well known). The authors may consider to further shorten such general descriptions and mainly highlight the new results obtained in this study.

Section 3.3.3: it is not clear whether the modeling analysis was conducted for the campaign average condition or for a particular case. Furthermore, the sub-title of this section may be not appropriate as this section only talked about the simulated OH level and O3 formation, other than the atmospheric oxidative capacity.

Page 21, Lines 8-10: based on the current analysis, I don't agree that the atmospheric oxidative capacity is stronger at the coastal WS than polluted TC site. HONO photolysis is a very important OH source in polluted areas including the TC site (Xue et al., 2016), which was not included in the present study. So the OH levels should be underestimated at TC. Moreover, the lower OH levels at TC should be due to the fast radical cycling given the more abundant VOCs. I presume that the HO2 and RO2 levels at TC should be significantly higher than those at WS.

Xue et al., Oxidative capacity and radical chemistry in the polluted atmosphere of Hong Kong and Pearl River Delta region: analysis of a severe photochemical smog episode. Atmospheric Chemistry and Physics. 16. 9891-9903. 2016.

Page 21, Lines 15-22: it is surprising that the O3 production rates at WS were much higher than those at TC, especially on the episode days, given that the NOx and VOC levels were much higher at TC than at WS during O3 episodes. What's the possible reason for this?

Section 3.3.3 and Figure 5: it would be better if the ozone production rates were expressed in ppb/h so that it can be easily compared with the observed ozone increase.

---

## Author Comment (AC1) · 28 Feb 2018

Responses to Referee #1

This paper provides a detailed study of trace gases and meteorology at two sites, one a suburban site in Hong Kong (labelled TC), the other a coastal site (labelled WS), with few local anthropogenic emissions, on the edge of the South China Sea. The sites are separated by ~ 40 km. Emphasis is given to ozone episodes (>100 ppbv) and near episodes, which occurred on a number of occasions, some extending over 9 days, during two ~50 day periods in August, September; October, November 2013. The results are rationalised in detail using a range of modelling techniques: a zero dimensional box model study using the master chemical mechanism (MCM); the Weather Research and Forecasting (WRF) model to provide wind fields and coupled with the CMAQ model to provide an Eulerian representation of the physical and chemical processes over a wide area and HYSPLIT to provide backward particle release simulations to understand air mass origins. The paper provides a very useful dataset and an interesting analysis.

Thanks for the positive comments and the concerns, which helped to improve the manuscript substantially. Responses are given item by item below following the specific comments, and revisions are made where necessary.

1. The results are discussed in terms of the interaction between marine and continental air. The episodic ozone concentrations are significantly higher at WS than at TC and much of the paper relates to a discussion of the origin of these differences, which are ascribed to weaker NO titration and to a stronger oxidative capacity at the coastal site. The main meteorological features during the episodes were tropical cyclones, with transport from the polluted Pearl River Delta Region to the sites, continental anticyclones, which again brought air from polluted inland areas and Sea Land Breezes, with alternation of onshore and offshore winds.

My main concern is with the contention that the results relate to the interaction between marine and continental air, which is included in the title and pervades the text. WS is one of several islands lying close to the coast. Its important characteristic is that there are few local emissions so that $NO_x$ is low. Other pollutants, CO, $SO_2$, NMHC show clear indications of advection of polluted air, but the concentrations are on average

lower than those found at TC. The wind patterns confirm that the air is primarily, perhaps exclusively during the episodes, of continental origin. Even the SLB winds from the sea simply advect high ozone concentrations, formed in polluted air, back to the coastal region. Marine air has much less impact than is found and has been widely discussed at, say, Mace Head in Ireland or Cape Grim in Tasmania.

The excellent comment is highly appreciated. As presented in the manuscript and summarized by the referee, this paper focused on the interaction between the continental and marine air in the coastal area of Hong Kong. The impacts of the continental air on air quality in marine boundary layer were discussed profoundly. Specifically, the polluted continental air masses were transported to the marine atmosphere under tropical cyclone, continental anticyclone and land breeze (section 3.2). As a result, the chemical compositions of the marine air changed substantially, leading to increased $O_3$ production under northerly winds at the reception of continental air (section 3.3). This process was further confirmed by the chemical transport model (section 3.4).

However, as concerned by the referee, we agree that the impacts of marine air on continental air quality were not discussed in such a comprehensive way. In fact, this effect was mainly described as sea breeze (section 3.2.3) and the intrusion of high $O_3$ formed over South China Sea into the continental area under sea breeze (section 3.4). In the revised manuscript, the alleviation of continental air pollution under oceanic flow is discussed, which represents a type of interaction between the continental and marine air. Furthermore, the enhancements of oceanic emission tracers (*e.g.* dimethyl sulfide) in the inland area under sea breeze are presented as an indication of marine influence. This is consistent with the findings at Mace Head in Ireland and Cape Grim in Tasmania. At last, what we want to emphasize is that this study focuses on $O_3$ pollution under the interaction between continental and marine air. The advection of marine air laden with $O_3$ back to the coastal areas is a typical interaction in this region, which is thought to be an important marine influence.

Revisions are made in the revised manuscript as follows.

The arrival of oceanic air masses generally brings substantial marine-originated compounds (e.g. dimethyl sulfide) to the continent and significantly alleviates the anthropogenic air pollution there. In fact, this is one of the main reasons for low $O_3$ mixing ratio observed in the PRD region in summertime when southwestern winds prevail (Wang et al., 2009; Wang et al., 2017a). In this study, it was also found that winds over the ocean increased the concentration of dimethyl sulfide at TC (see Figure S8) and reduced the levels of almost all man-made air pollutants in many cases, mainly in summertime (Figure 2a). In contrast, sea breezes carrying elevated $O_3$ formed over SCS might build up the terrestrial $O_3$ in the coastal area in some cases.

[Figure]

**Figure S8** Average concentrations of dimethyl sulfide (DMS) observed at TC and WS when continental or marine air masses dominate.

For details, please refer to Page 29, Lines 2-11 in the revised manuscript and in the supplement.

2. The observation of higher ozone at WS compared with TC derives primarily, as argued, from the low emissions at WS and the consequently much lower $NO_x$ and reduced titration via NO + $O_3$. The most telling observation is the near equivalence of

the total oxidant concentration at the two sites during both episodes and non-episodes (p 20). Similar behaviour is of course found in many other locations when comparing rural and urban ozone concentrations in similar air masses. It is the absence of local $NO_x$ emissions at WS that leads to the differences; it is not specifically related to its coastal location and certainly not to marine influences.

Thanks for the comments. We agree that, due to the influence of NO titration, the near equivalence of the total oxidant concentration is common between two sites where primary air pollutants are intensively emitted at one site and then transported to the other site. Indeed, the similar behavior at WS and TC was partially attributable to this effect (see section 3.3.2). However, this study also demonstrated that the more intensive in-situ $O_3$ formation due to the stronger oxidative capacity of the atmosphere could be another important factor for the higher $O_3$ at WS (see section 3.3.3). Additionally, $O_3$ formed during the transport of polluted air masses from the continent to the marine atmosphere might also elevate $O_3$ at WS, which will be discussed in a companion paper (Wang et al., 2018). In fact, wherever the observed $O_3$ was formed, *i.e.* at WS or during the transport of air masses from the inland area to WS, the high $O_3$ at WS was a reflection of the interaction between the continental and marine air, because WS was nearly free of anthropogenic emissions where $O_3$ and its precursors were originated from the continent. Certainly, this interaction was mainly manifested as the impacts of continental air on marine air quality. We understood that the marine influences should be emphasized as a part of interaction between the continental and marine air. As responded to the previous comment (comment #1), discussions on the marine influences are extended in the revised manuscript.

3. The discussion of the daily ozone profile could also be improved. The diurnal variation is superimposed on a residual night-time ozone concentration, which is substantial, Figs 2 and S4. This might be discussed.

Many thanks for the comment. The suggested discussion has been provided in the revised manuscript.

In this study, the average NO mixing ratio at TC was 14.0$\pm$0.8 ppbv, compared to 0.7$\pm$0.1 ppbv at WS (Table 1). The much lower NO at WS implied weaker titration to $O_3$, which enabled the survival of more $O_3$ and caused substantial residual $O_3$ at WS particularly at night time when there were no photochemical reactions (Figure 2 and Figure S6).

For details, please refer to Page 21, Lines 9-12.

4. Is the higher rate of ozone formation, shown in Fig 5b, a reflection of the high ozone concentration itself? It would be helpful to show the concentrations of OH, $HO_2$ and $RO_2$ vs time and also their rates of production and loss. Is the enhanced [OH] a result of increased $O^1D$ production from the higher [$O_3$] found at WS in episodes? These plots could, if necessary, be shown in the Supplement.

We are grateful for the good comments. The higher net $O_3$ production during $O_3$ episodes at WS, as shown in Figure 5 (b) of the original manuscript, was directly caused by the enhanced reaction rates of $RO_2$+NO and $HO_2$+NO. This was associated with the increase of $O_3$ precursors, particularly $NO_x$, during $O_3$ episodes. Under the assumption that local $O_3$ formation dominated $O_3$ budget at WS, the higher $O_3$ production rate during $O_3$ episodes resulted in higher $O_3$, or it was a reflection of the higher $O_3$. However, the higher $O_3$ production rate was not caused by the higher observed $O_3$ during episodes, which were not input into the model.

As suggested, the concentrations of OH, $HO_2$ and $RO_2$, as well as their production and loss rates, are presented in the revised supplement with discussions in the revised manuscript. The enhanced [OH] during $O_3$ episodes cannot be totally attributable to the increased $O^1D$ ($O_3$ photolysis). Instead, most of the OH increase was attributable to the enhanced reaction rate between $HO_2$ and NO during $O_3$ episodes.

[revised manuscript text omitted]

For details, please refer to Section 3.3.3 (Page 22) and Figure 5 in the revised manuscript and Figure S7 in the revised supplement.

5. It would also be helpful, again in the Supplement, to see ozone, OH, $HO_2$ and $RO_2$ concentrations, and ozone and radical rates of formation and loss on a specific episode day. Using averages can lead to a loss of clarity and understanding.

Thanks for the suggestion. The concentrations and formation/loss rates of radicals, *i.e.* OH, $HO_2$ and $RO_2$, and $O_3$ are provided on daily basis in the revised manuscript and revised supplement. More discussions are given for better understanding of the photochemistry.

For details, please refer to the responses to comment #4.

6. Two additional points: $O_1D$ in the caption to Fig 5 should be $O^1D$. The English needs a good deal of attention, particularly the frequent absence of definite / indefinite articles.

Sorry for the mistake in $O_1D$, which is corrected to $O^1D$ throughout the manuscript. The English, particularly the absence of definite/indefinite articles, has been double checked and revised where necessary.

7. The paper makes a substantial contribution and should be published in ACP. The authors, though, should consider the points made above relating to the overall emphasis of the paper and the clarity of the discussion on chemical processes.

Thanks again for the positive comments on the paper. Revisions are made according to the comments and suggestions, which mainly include the discussions on marine influences and chemical processes at both sites. We hope that the revised manuscript is satisfactory to the referee.

Responses to Referee #2

This paper reports intensive field measurements at two sites over the South China Sea. The spatial distribution of ozone pollution and its favorable synoptic conditions were interpreted. The authors also tried to link, by the sea-land breeze, the transport of continental pollution to oceans and the recirculation of land-originated aged air masses from ocean to the coastal regions. The manuscript is generally well written and easy to follow. The following specific comments should be addressed before it can be considered for publication at ACP.

Specific Comments:

1. Page 1, Line 6: Hong Kong, China.

Thanks for the comment. The author's address has been amended.

2. Page 2, Lines 5-6: the authors may either spell out TC and WS, or just remove them from the abstract.

Thanks for the suggestion. The sampling sites (i.e. TC and WS) have been spelled out.

3. Page 2, Line 15: the word "magnified" may be not appropriate here. The ozone-laden air may be transported to a larger area over the oceans, but should not be "magnified".

Thanks for the Referee's concern. Since this study successfully demonstrates $O_3$ concentration and production increased at the marine site due to the strong atmospheric oxidative capacity as well as the changed chemical compositions at the reception of continental air, the authors think that the use of "magnified" here is not improper.

4. Page 3, Lines 14-19: to date, the long-term $O_3$ trend studies were relatively limited in

China. The authors should refer to the following earlier studies in Hong Kong, the PRD region and northern China.

Sun, et al., Significant increase of summertime ozone at Mount Tai in Central Eastern China, Atmos. Chem. Phys., 16, 10637-10650, 2016.

Xue et al., Increasing external effects negate local efforts to control ozone air pollution: a case study of Hong Kong and implications for other Chinese cities, Environ. Sci. Tech., 48, 10769-10775, 2014.

Thanks for providing these references. They have been cited in the revised manuscript.

"However, increasing studies showed that surface $O_3$ was elevated rapidly in East Asia in the last decade (Ding et al., 2008; Xu et al., 2008; Parrish et al., 2012; Xue et al., 2014; Zhang et al., 2014; Sun et al., 2016; Lin et al., 2017; Wang et al., 2017a). For example, the observational data revealed that regional $O_3$ concentrations increased at a rate of 0.86 ppbv $yr^{-1}$ in Pearl River Delta (PRD) from 2006 to 2011 (Li et al., 2014), at a rate of 0.56 ppbv $yr^{-1}$ in Hong Kong from 2005 to 2014 (Wang et al., 2017a), and even at a rate of 1.7-2.1 ppbv $yr^{-1}$ (summertime only) at Mount Tai in central eastern China (Sun et al., 2016)."

For details, please refer to page 3, lines 13-20.

5. Page 4, Lines 24-26: a more recent study has investigated the detailed chemical features including the radical chemistry in different air masses arriving at the South China Sea.

Li et al., Oxidizing capacity of the rural atmosphere in Hong Kong, Southern China. Science of the Total Environment. 612. 1114-1122. 2018.

Thanks for providing this new reference. Li et al. (2018) has been cited in the revised manuscript.

So far, only a handful of studies deeply evaluated the chemical characteristics of air masses under various synoptic systems (Wang et al., 2005; Guo et al., 2009; Guo et al., 2013; Li et al., 2018).

For details, please refer to Page 4, Lines 26-28.

6. Page 5, Lines 21-22: it is not the case that northeast monsoon prevails in late summer. The $O_3$ episode occurring in late summer in Hong Kong is mainly related to the tropical cyclones.

Yes, tropical cyclones are one of the principal factors leading to $O_3$ episodes occurring in late summer and autumn in Hong Kong, which has been acknowledged in the Section 3.2 in the original manuscript. In addition, northeast monsoon prevailing in autumn would also contribute to high $O_3$ mixing ratio observed in Hong Kong by bringing in high concentrations of $O_3$ and its precursors from the PRD and other heavy-polluted areas. To make this point clearer, the sentence has been revised and now reads as follows:

High $O_3$ mixing ratios are frequently observed in Hong Kong in late summer and autumn (Ling et al., 2013) when tropical cyclones and the northeast monsoon prevail, respectively.

For details, please refer to Page 5, Lines 27-28.

7. Page 6, Lines 8-10: it has been known that the traditional commercial $NO_x$ analyzer may be subject to significant positive interference for the $NO_2$ measurements, especially at the rural and remote areas like WS. The authors need state the uncertainty of the $NO_2$ measurements and the subsequent observation-based modeling analysis.

Thanks for the comment. The uncertainty of the $NO_x$ measurements has been added in the method section.

It was noteworthy that the measured $NO_x$ might include other oxidized reactive nitrogen that was converted by the molybdenum. Thus, the $NO_x$ concentrations given below

were considered the upper limits of their actual values (Dunlea et al., 2007; Ran et al., 2011).

In addition, the inherent uncertainty of $NO_x$ measurement mentioned above might slightly affect the modeling results.

For details, please refer to Page 6, Lines 12-15 and Page 9, Lines 9-10.

8. Page 8, Lines 16-18: so the OBM was not constrained by the measured HONO and OVOCs, right? This may affect the accurate modeling of OH radicals and ozone formation. As the OVOC measurements were available in the present study, the authors should constrain the model with the measured OVOC data.

Thanks for the comment. The measured OVOCs, as well as the HONO obtained from previously published data, have been used to constrain the OBM. The new results are presented in the revised manuscript. Please see Comment 13 for more details.

9. Page 9, Lines 19-27: it would be better if the authors could provide the time series of model simulations and observations for a direct comparison, maybe in the supporting information.

Thanks for the good suggestion. For comparison, the time series of model simulations and observations have been added in Figure S3 in the revised supplement.

[Figure]

**Figure S3.** Time series of the WRF-CMAQ simulated and the observed CO and $O_3$ at WS (left panel) and TC (right panel) during a typical $O_3$ episode on Oct. 2-4, 2013.

10. Page 10, Table 1: it would be much better if the statistics of the most abundant NHMC and carbonyl species are individually shown, instead of the bulk concentrations.

Thanks for the suggestion. The statistics of the top 10 NHMC and the top 3 carbonyl species are individually shown in Table S5 in the revised supplement.

**Table S5**. Statistics (Mean ± 95% C.I.) of the top 10 NMHC and the top 3 carbonyl species observed at TC and WS during $O_3$ episodes and non- episodes (unit: pptv).

| Compound | TC | | WS | |
|---|---|---|---|---|
| | *Episode* | *Non-episode* | *Episode* | *Non-episode* |
| *Ethane* | 2179±222 | 1852±256 | 2077±182 | 1456±167 |

| | | | |
|---|---|---|---|
| *Propane* | 1966±277 | 1572±207 | 1523±126 | 866±126 |
| *i-Butane* | 1944±371 | 1433±166 | 1559±167 | 810±115 |
| *Acetylene* | 2083±165 | 1316±145 | 1805±133 | 1086±122 |
| *Toluene* | 1829±365 | 1357±254 | 1737±388 | 703±183 |
| *n-Butane* | 1437±163 | 1336±148 | 1160±145 | 480±113 |
| *n-Hexane* | 733±329 | 1351±443 | 980±299 | 447±121 |
| *Ethene* | 1140±167 | 1077±171 | 826±99 | 691±94 |
| *i-Pentane* | 964±145 | 813±123 | 918±99 | 523±96 |
| *Benzene* | 614±49 | 428±51 | 587±47 | 381±44 |
| *Formaldehyde* | 5068±454 | 3522±286 | 4257±355 | 2471±180 |
| *Acetone* | 5064±831 | 3367±445 | 3984±287 | 2086±162 |
| *Acetaldehyde* | 1807±162 | 1241±115 | 1618±133 | 920±105 |

11. Section 3.2: this section is too long and contains a lot of general description of the typhoon, continental anticyclone, and sea-land breeze (most of them are already well known). The authors may consider to further shorten such general descriptions and mainly highlight the new results obtained in this study.

Thanks for the comment. The Section 3.2 has been further shortened in the revised manuscript by removing some simple descriptive text, for example:

"The main feature of the anticyclones is sinking air at the center with gentle clockwise winds in the northern hemisphere. The air warms up as it sinks by compression leading to warm, cloudless and dry weather, which is conducive to intensive photochemical O3 formation. In addition, anticyclone is a large-scale weather system which produces long-lasting settled and calm weather for many days or weeks favorable to the accumulation of primary and secondary pollutants." and

"In general, the temperature difference between the sea and the land is large on the SLB days. Taking 3 Oct. as an example, the maximum hourly temperature at TC was

3.2 $^{o}$C higher than that at WS during daytime hours, whereas the minimum hourly temperature in the evening was 2.7 $^{o}$C lower at TC than at WS."

For details, please refer to Section 3.2 (Pages 14-19) in the revised manuscript.

12. Section 3.3.3: it is not clear whether the modeling analysis was conducted for the campaign average condition or for a particular case. Furthermore, the sub-title of this section may be not appropriate as this section only talked about the simulated OH level and O3 formation, other than the atmospheric oxidative capacity.

Thanks for the comment. The modeling analysis in section 3.3.3 was conducted for the individual days when VOCs were collected. For the subtitle, we have discussed the atmospheric oxidative capacity in the revised manuscript according to the definition in Elshorbany et al. (2009) and Xue et al. (2016), *i.e.* oxidation rate of VOCs by OH. However, since $O_3$ production rate is also an important content in this section, the subtitle has been changed to "Atmospheric oxidative capacity and $O_3$ production rate".

*3.3.3 Atmospheric oxidative capacity and O₃ production rate*

$O_3$ formation is driven by the transformation and recycling of oxidative radicals, including OH, $HO_2$ and $RO_2$, collectively referred to as $RO_x$ hereafter. The production and loss rates of these radicals, and their equilibrium concentrations on the canister sampling days were simulated by the PBM-MCM model, as shown in Figure S7.

For details, please refer to Page 22, Lines 20-24.

The overall oxidation rate of VOCs by OH was employed to indicate the atmospheric oxidative capacity in previous studies (Elshorbany et al., 2009; Xue et al., 2016). In this study, we found that the oxidation rate of VOCs at TC ($6.1\pm2.1\times10^6$ molecules cm$^{-3}$ s$^{-1}$ during $O_3$ episodes and $5.7\pm0.9\times10^6$ molecules cm$^{-3}$ s$^{-1}$ during non-episodes) was remarkably ($p<0.05$) lower than that at WS ($O_3$ episode: $15\pm2.5\times10^6$ molecules cm$^{-3}$ s$^{-1}$ and non-episode: $8.9\pm1.3\times10^6$ molecules cm$^{-3}$ s$^{-1}$). The results revealed that the atmospheric oxidative capacity at TC was weaker than at WS for both $O_3$ episodes and non-episodes, inconsistent with the findings of Elshorbany et al. (2009) and Xue et al. (2016) who concluded that the atmospheric oxidative capacity was higher in more

polluted environments due to the fact that the atmospheric oxidative capacity is positively proportional to the VOCs and OH levels. Both Elshorbany et al. (2009) and Xue et al. (2016) reported very high mixing ratios of VOCs (*e.g.* toluene of 9.5 and 6.3 ppbv, respectively) in the polluted cases, which explained the strong atmospheric oxidative capacity. However, in this study, it is more likely that the higher $NO_x$ at TC consumed more OH and resulted in lower oxidative capacity than at WS, despite the slightly higher VOCs at TC (Table 3).

For details, please refer to Page 23, Lines 20-33, and Page 24, lines 1-2.

13. Page 21, Lines 8-10: based on the current analysis, I don't agree that the atmospheric oxidative capacity is stronger at the coastal WS than polluted TC site. HONO photolysis is a very important OH source in polluted areas including the TC site (Xue et al., 2016), which was not included in the present study. So the OH levels should be underestimated at TC. Moreover, the lower OH levels at TC should be due to the fast radical cycling given the more abundant VOCs. I presume that the $HO_2$ and $RO_2$ levels at TC should be significantly higher than those at WS.

Xue et al., Oxidative capacity and radical chemistry in the polluted atmosphere of Hong Kong and Pearl River Delta region: analysis of a severe photochemical smog episode. Atmospheric Chemistry and Physics. 16. 9891-9903. 2016.

The excellent comment is highly appreciated. Firstly, HONO was indeed not input into the model for simulation of photochemistry in this study, as we did not measure HONO concentrations in the sampling campaign. We agree that the absence of HONO might have an influence on the conclusions about photochemistry at the two sites. Therefore, the average diurnal profiles of HONO observed at TC and a coastal background site (Hok Tsui, HT) in Hong Kong were used to present the average levels of HONO at TC and WS, respectively, for model simulations again.

HONO has been recognized as an important source of OH, influencing $O_3$ formation significantly (Kleffmann, 2007). Since we did not measure HONO mixing ratios in this study, the average diurnal profiles of HONO observed at TC in autumn 2011 (Xu et al., 2015) and at a coastal background site (Hok Tsui, HT) in southeast Hong Kong in autumn 2012 (Zha, 2015) were applied to the photochemical simulations at TC and WS, respectively. Figure S1 shows the

average diurnal cycles of HONO at TC and HT. The use of the aforementioned diurnal profiles might increase the uncertainty of model simulation. However, we believe that the newly introduced uncertainties could not be too high, because HONO observations at TC and HT were carried out 2 years and 1 year before the sampling campaign of this study, respectively. In addition, HT was comparable to WS in aspects of local emissions (nearly free of anthropogenic emissions), air mass category (mixed continental and marine air) and location (to the south of Hong Kong and on SCS).

For details, please refer to Section 2.3 (Pages 8-9).

However, with the inclusion of HONO, the simulated OH, $HO_2$ and $RO_2$ at TC were still lower than those at WS. To keep consistency with previous studies (Elshorbany et al., 2009; Xue et al. 2016), the atmospheric oxidative capacity is defined as the overall oxidation rate of VOCs by OH in the revised manuscript. We found that the atmospheric oxidative capacity was also much higher at WS than at TC, due to the higher OH concentration at WS despite the lower VOC levels. In fact, according to our analyses, the lower OH at TC was more related to the higher $NO_2$, which served as a scavenger of OH through the formation of $HNO_3$. The lower $HO_2$ and $RO_2$ at TC was possibly resulted from their conversion to OH and RO ($HO_2$ and OH subsequently), under the condition of sufficient NO. However, the recycled OH could be further removed by reacting with $NO_2$. As a consequence, OH, $HO_2$ and $RO_2$ were progressively consumed, which caused their lower concentrations at TC. This section has been substantially revised as follows.

[revised manuscript text omitted]

For details, please refer to Section 3.3.3 in the revised manuscript.

14. Page 21, Lines 15-22: it is surprising that the $O_3$ production rates at WS were much higher than those at TC, especially on the episode days, given that the NOx and VOC

levels were much higher at TC than at WS during O3 episodes. What's the possible reason for this?

Thanks for the comment and question. With the addition of HONO in $O_3$ simulation, the new modeling results show that the net $O_3$ production rate at WS was comparable to that at TC during non-episodes. However, it was much higher than that at TC during $O_3$ episodes, due to the more abundant peroxy radicals ($RO_2$ and $HO_2$) at WS, in addition to the increased NO during $O_3$ episodes which unleashed the potential of $O_3$ production through the reactions between peroxy radicals and NO. It should be noted that the increase of NO at WS during $O_3$ episodes did not lead to $O_3$ reduction, unlike the situation in most urban environments including TC, because $O_3$ formation at WS was limited by both VOCs and $NO_x$ and more sensitive to $NO_x$ without the input of continental air. Overall, despite the lower VOCs and $NO_x$, the concentrations of peroxy radicals at WS were higher than at TC (the reasons have been discussed in responses to comment #13), and the increase of NO during $O_3$ episodes accelerated $O_3$ formation through the reactions between peroxy radicals and NO.

Furthermore, the production and loss rates of $O_3$ were simulated (Figure 5(b)). Despite the increased $O_3$ mixing ratio during episodes (Table 3), there was no significant change in net $O_3$ production between $O_3$ episodes (2.5±1.0 ppbv/h) and non-episodes (2.5±0.5 ppbv/h) at TC ($p$>0.05), suggesting that regional transport might play critical roles in regulating $O_3$ levels at TC. In fact, previous studies (Huang et al., 2006; Jiang et al., 2008) have repeatedly confirmed that $O_3$ pollution at this site could be aggravated under northerly winds and/or downdraft on the periphery of typhoon. In contrast, the net $O_3$ production increased remarkably from non-episodes (2.8±0.5 ppbv/h) to $O_3$ episodes (6.6±1.1 ppbv/h) at WS. Obviously, $O_3$ production at WS was much higher than at TC during $O_3$ episodes, while they were comparable during non-episodes. This was likely due to the more abundant peroxy radicals ($RO_2$ and $HO_2$) at WS than at TC, in addition to the increased $NO_x$ during $O_3$ episodes which enhanced the reactions between the peroxy radicals and NO (increasing $O_3$ formation). Insight into the $O_3$ production pathways found that the reaction rates of $RO_2$+NO and $HO_2$+NO were significantly enhanced from 1.6±0.2 and 2.0±0.4 ppbv/h during non-episodes to 3.2±0.5 and 5.2±0.9 ppbv/h during $O_3$ episodes, respectively. Our recent study (Wang et al., 2017b) revealed that $O_3$ formation at WS was in a

transition regime and much more sensitive to $NO_x$ during non-episodes, when $O_3$ production through peroxy radicals reacting with NO was seriously limited by the low $NO_x$. During $O_3$ episodes, with the increase $O_3$ precursors (particularly $NO_x$), these reactions were accelerated and the net $O_3$ production increased substantially. Detailed discussion on the $O_3$ photochemistry at WS can be found in our recent publication (Wang et al., 2017b).

For details, please refer to Section 3.3.3 in the revised manuscript.

15. Section 3.3.3 and Figure 5: it would be better if the ozone production rates were expressed in ppb/h so that it can be easily compared with the observed ozone increase.

The good suggestion has been accepted with thanks.